# EHD2 overexpression promotes tumorigenesis and metastasis in triple-negative breast cancer by regulating store-operated calcium entry

Haitao Luan[1†], Timothy A Bielecki[1†], Bhopal C Mohapatra[2,3†], Namista Islam[1,2], Insha Mushtaq[1,4], Aaqib M Bhat[1,2], Sameer Mirza[2], Sukanya Chakraborty[1,2], Mohsin Raza[2], Matthew D Storck[1], Michael S Toss[5], Jane L Meza[3,6], Wallace B Thoreson[7], Donald W Coulter[3,8], Emad A Rakha[5], Vimla Band[2,3]*, Hamid Band[1,2,3,4]*

[1]Eppley Institute for Research in Cancer and Allied Diseases, University of Nebraska Medical Center, Omaha, United States; [2]Department of Genetics, Cell Biology and Anatomy, College of Medicine, University of Nebraska Medical Center, Omaha, United States; [3]Fred & Pamela Buffett Cancer Center, University of Nebraska Medical Center, Omaha, United States; [4]Department of Pathology & Microbiology, College of Medicine, University of Nebraska Medical Center, Omaha, United States; [5]Department of Histopathology, Nottingham University Hospital NHS Trust, City Hospital Campus, Nottingham, United Kingdom; [6]Department of Biostatistics, College of Public Health, University of Nebraska Medical Center, Omaha, United States; [7]Stanley M. Truhlsen Eye Institute, University of Nebraska Medical Center, Omaha, United States; [8]Department of Pediatrics, University of Nebraska Medical Center, Omaha, United States

*For correspondence:
vband@unmc.edu (VB);
hband@unmc.edu (HB)

†These authors contributed equally to this work

**Abstract** With nearly all cancer deaths a result of metastasis, elucidating novel pro-metastatic cellular adaptations could provide new therapeutic targets. Here, we show that overexpression of the EPS15-Homology Domain-containing 2 (EHD2) protein in a large subset of breast cancers (BCs), especially the triple-negative (TNBC) and HER2+ subtypes, correlates with shorter patient survival. The mRNAs for EHD2 and Caveolin-1/2, structural components of caveolae, show co-overexpression across breast tumors, predicting shorter survival in basal-like BC. *EHD2* shRNA knockdown and CRISPR-Cas9 knockout with mouse *Ehd2* rescue, in TNBC cell line models demonstrate a major positive role of EHD2 in promoting tumorigenesis and metastasis. Mechanistically, we link these roles of EHD2 to store-operated calcium entry (SOCE), with EHD2-dependent stabilization of plasma membrane caveolae ensuring high cell surface expression of the SOCE-linked calcium channel Orai1. The novel EHD2-SOCE oncogenic axis represents a potential therapeutic target in EHD2- and CAV1/2-overexpressing BC.

## Editor's evaluation

This paper reports important findings regarding the role of EHD2 over-expression in a subset of poor-prognosis breast cancers. Through analysis of extensive patient data and in vitro and xeno-graft experiments with cell-line models, the paper provides solid evidence that EHD2, a component of caveolae, has strong pro-tumorigenic and pro-metastatic roles in these cancers by regulating calcium channels. The paper should be of interest to investigators studying metastasis and the role of caveolae in calcium signaling and homeostasis.

## Introduction

Breast cancer (BC) remains a major cause of cancer-related deaths, with less than 30% 5-year survival rate in patients with metastatic disease (https://www.acs.org/). Triple-negative BC (TNBC) presents a particularly difficult diagnosis with lack of targeted therapies. A better understanding of tumorigenesis- and metastasis-associated cellular adaptations could open novel approaches to improve the survival of metastatic BC patients.

EPS15-homology (EH) domain-containing (EHD) proteins (EHD1-4) are evolutionarily conserved lipid membrane-activated ATPases that regulate inward or outward vesicular traffic between the plasma membrane and intracellular organelles by controlling tubulation and scission of trafficking vesicles (*Naslavsky and Caplan, 2011*). Unlike other family members, which predominantly localize to endosomal and other intracellular compartments, EHD2 is known to primarily localizes to plasma membrane caveolae to maintain their stable membrane pool (*Morén et al., 2012*; *Stoeber et al., 2012*), suggesting a likely role in caveolae-associated cellular functions. Indeed, caveolae-dependent fatty acid uptake in adipocytes and eNOS-NO induced small blood vessel relaxation are impaired in *Ehd2* knockout mice (*Matthaeus et al., 2020*; *Matthaeus et al., 2019*). EHD2-dependent stabilization of caveolae was also found to promote the cell surface expression of ATP-sensitive $K^+$ channels and protect cardiomyocytes against ischemic injury (*Yang et al., 2018*). Caveolae are key to buffering the plasma membrane stress (*Sinha et al., 2011*) and EHD2 has been shown to positively regulate mechano-transduction through re-localization to the nucleus and regulation of gene transcription (*Torrino et al., 2018*).

Recent studies have painted a complex picture of the potential roles of EHD2 in cancer. Reduced EHD2 expression was reported in esophageal, colorectal, breast, and hepatocellular cancers (*Li et al., 2013*; *Guan et al., 2021*; *Yang et al., 2015*; *Liu et al., 2016*), with in vitro knockdown or overexpression studies supporting a tumor suppressive role for EHD2. On the contrary, EHD2 overexpression was found as a component of a mesenchymal signature in malignant gliomas with shorter survival, and knockdown analyses showed the EHD2 requirement for cell proliferation, migration, and invasion (*Zhang et al., 2021*). Higher *EHD2* mRNA expression in papillary thyroid carcinomas was associated with extrathyroidal extension, lymph node metastasis, higher risk of recurrence, and presence of BRAF-V600E mutation (*Kim et al., 2017*). Studies of clear cell renal cell carcinoma also supported a positive role of EHD2 in tumorigenesis (*Liu et al., 2019*). A recent study provided a more mixed picture, with loss of EHD2 expression in TNBC cell lines enhancing their proliferation, migration, and invasion but low levels of *EHD2* mRNA in TNBC patient tumors predicting better prognosis (*Shen et al., 2020*). Thus, a definitive role of EHD2 in oncogenesis and its mechanisms remain unclear.

Here, our comprehensive expression analyses in BC samples and in vitro and in vivo studies using *EHD2* knockdown or knockout approaches in TNBC cell models provide definitive evidence for strong pro-tumorigenic and pro-metastatic role of EHD2. Our studies suggest a novel pro-oncogenic mechanism of EHD2, namely its requirement for efficient store-operated calcium entry (SOCE), a pathway known to promote tumorigenesis and metastasis in breast and other cancers (*Mo and Yang, 2018*; *Yang et al., 2009*).

## Results

### EHD2 is expressed in basal cells of the mouse mammary gland and in a subset of basal-like breast cancer cell lines

First, we used immunoblotting and immunofluorescence staining of mammary gland tissue from control and *Ehd2*-null mice (generated in the lab; unpublished) to authenticate the specific recognition of EHD2 by an antibody previously validated against ectopic tagged EHD2 (*George et al., 2007*; *Figure 1A–B*). High EHD2 expression was seen in mammary adipocytes, consistent with high EHD2 expression in adipose tissues (*Morén et al., 2019*). Moderate/high EHD2 staining was seen in the mammary basal/myo-epithelium (smooth muscle actin$^+$), but little in the luminal epithelium (cytokeratin 8$^+$; *Figure 1C*). The basal/myoepithelial cell selective localization was confirmed by immunohistochemistry (IHC; *Figure 1D*). Immunoblotting of basal (EPCAM-low/CD29-high) and luminal (EPCAM-high/CD29-low) mouse mammary epithelial cell-derived organoids further confirmed the basal cell expression of EHD2 (*Figure 1E*). Thus, while mammary adipocytes express the highest levels of EHD2, within the epithelium the basal epithelial cells selectively express higher EHD2 levels.

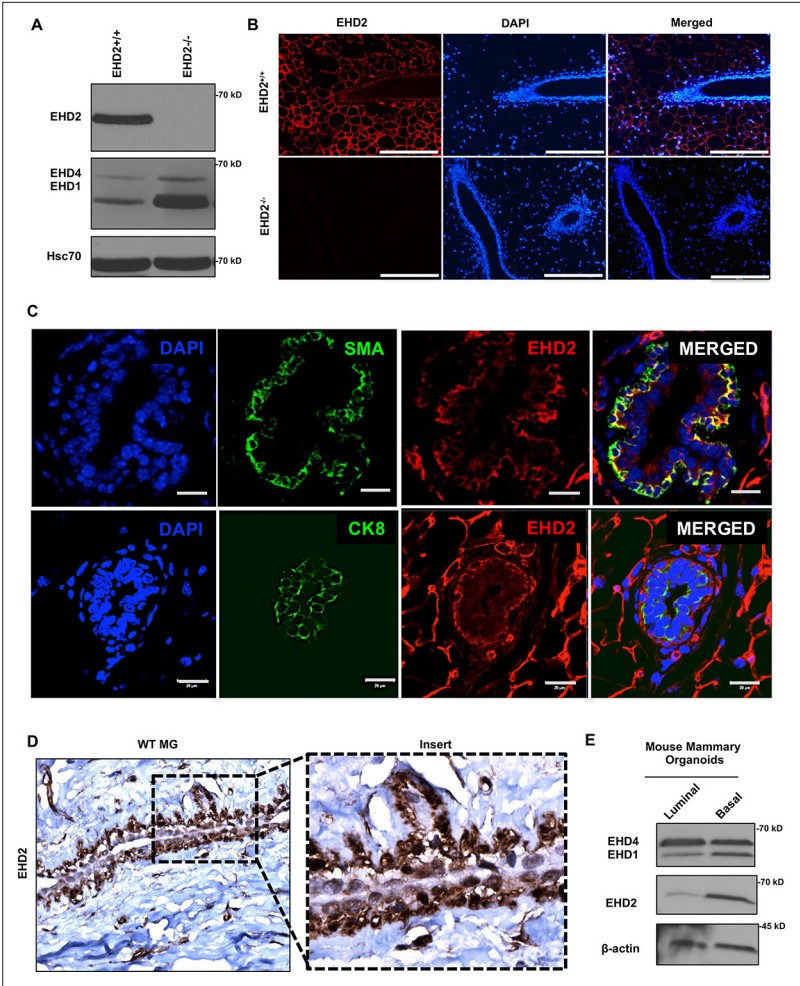

**Figure 1.** EHD2 is expressed in basal-like mammary epithelial cells. (**A and B**) Immunoblot (A) and immunofluorescence (B; scale bar, 20 µm) analysis of wildtype (*Ehd2*+/+) and *Ehd2*-null (*Ehd2*-/-) mouse mammary gland to validate the specific reactivity of anti-EHD2 antibody used in this study. (**C**) Immunofluorescence analysis of EHD2 expression in basal vs. luminal epithelial cells of normal mouse mammary gland. Top panel, EHD2 (red) co-staining with basal cell marker alpha smooth muscle actin (SMA; green); Bottom panel, EHD2 (red) co-staining with luminal cell marker cytokeratin 8 (CK8; green). Nuclei are stained with DAPI (blue). Scale bars, 20 µm. (**D**) Confirmation of the basal epithelial cell-selective EHD2 expression in mouse mammary gland by immunohistochemical staining. Magnification, 200X. (**E**) Predominant basal epithelial cell expression of EHD2 revealed by immunoblot analysis of Matrigel-grown organoids derived from FACS-sorted EPCAM-low/CD29-high (basal) vs. EPCAM-high/CD29-low (luminal) mouse mammary epithelial cell populations.

The online version of this article includes the following source data for figure 1:

**Source data 1.** Original blots of *Figure 1A*.

**Source data 2.** Original blots of *Figure 1E*.

By immunoblotting, we found EHD2 expression in immortal basal-like mammary epithelial cell lines 76Ntert (hTert-immortalized primary mammary epithelial cell line; *Zhao et al., 2010*) and MCF10A, in 2 out of 3 TNBC cell lines, and at lower levels in 3 out of 11 HER2 + cell lines, but in none of the 9 luminal A/B BC cell lines (*Figure 2A*). Immunofluorescence analysis of selected cell lines confirmed the expression pattern seen in immunoblotting and showed exclusive localization of EHD2 to the plasma membrane and cytoplasm (*Figure 2B*). Notably, our cell line results were discordant with reports of comparable EHD2 expression in MCF-7 (luminal), MDA-MB-415 (luminal), and MDA-MB-231 (basal) cell lines (*Yang et al., 2015*; *Shi et al., 2015*). The pattern of EHD2 protein expression we observed correlated with the EHD2 mRNA expression data in the CCLE database (*Dai et al., 2017*; *Figure 2C*). Furthermore, by relating the CCLE data to the reported BC cell line subtype analysis (*Lehmann et al.,*

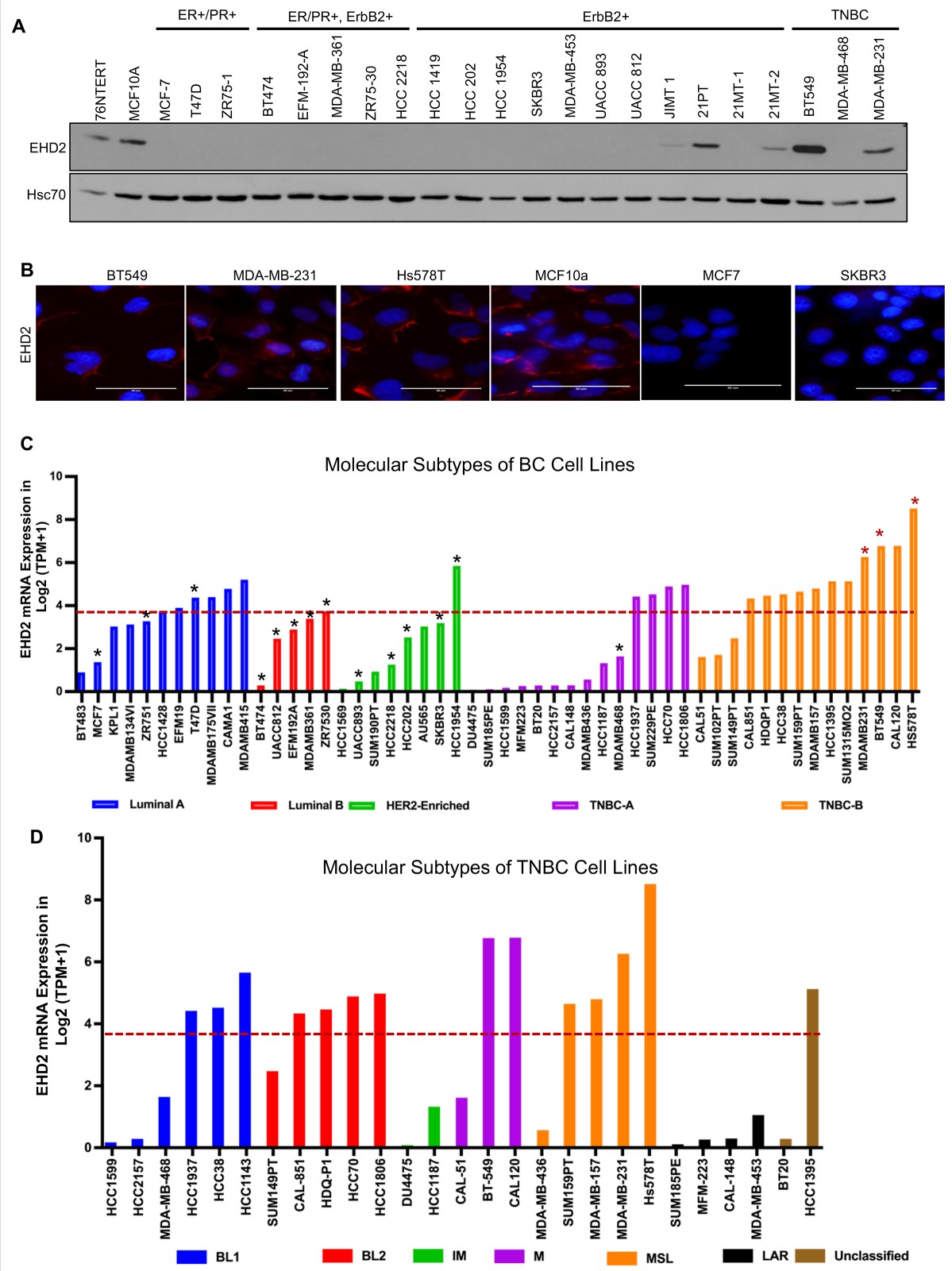

**Figure 2.** High EHD2 expression is preferentially observed in basal-like immortal mammary epithelial cell lines and in triple-negtaive/basal and HER2+ breast cancer cell lines. (**A**) Immunoblot analysis of EHD2 expression in non-tumorigenic immortal basal-like (76Ntert, MCF10a), and luminal A (ER+/PR+), luminal B (ER+/PR+, ErbB2+), ErbB2+, and Triple-negative (TN) breast cancer cell lines. (**B**) Immunofluorescence microscopy analysis of selected cell lines from A to further validate EHD2 (Red) expression pattern, showing predominant cytoplasmic and membrane localization. DAPI (blue) marks the

*Figure 2 continued on next page*

*Figure 2 continued*

nuclei. Scale bar, 50 μm. (**C**) EHD2 mRNA expression in breast cancer cell lines corresponding to major molecular subtypes as described in *Dai et al., 2017*. (**D**) EHD2 mRNA expression in breast cancer cell lines corresponding to TNBC subtypes as described in *Lehmann et al., 2011*. In C and D, The CCLE mRNA expression data is obtained as follows (per the CCLE site): RNASeq files are aligned with STAR and quantified with RSEM, and then TPM normalized. Reported values are Log2 (TPM + 1); TPM, transcripts per million. The dotted line represents the median expression value of EHD2 among all (N=63) BC cell lines. The black and red asterisks (*) in C indicate cell lines that we show as negative or positive for EHD2 expression by western blotting or immunofluorescence microscopy (**A** and **B**).

The online version of this article includes the following source data for figure 2:

**Source data 1.** Original blots of *Figure 2A*.

*2011*), we found higher *EHD2* mRNA expression to be a feature of TNBC cell lines, especially the mesenchymal-type (*Figure 2D*), similar to our western blot results (*Figure 2A*) where both EHD2-expressing TNBC cell lines were of the mesenchymal-type *Figure 2D*. Thus, high EHD2 expression is a feature of normal basal mammary epithelial cells and a subset of the basal-like/triple-negative BC.

## EHD2 overexpression is associated with metastasis and shorter survival in breast cancer

Based on the above findings, we conducted IHC staining of a tissue microarray (TMA) with 840 primary BC samples from a well-annotated patient cohort (*Abd El-Rehim et al., 2005*) to assess the expression of EHD2. Given the predominantly cytoplasmic/membrane localization of EHD2 in the mammary gland and BC cell lines, but the reported nuclear localization in cell lines under defined conditions (*Torrino et al., 2018*; *Pekar et al., 2012*), we quantified the IHC signals as cytoplasmic and nuclear (*Figure 3A*). 759 and 756 cases respectively showed a valid positive/negative cytoplasmic or nuclear signal (*Supplementary file 1*-Table 1A). High cytoplasmic and low nuclear EHD2 signals showed a positive association with higher tumor grade, higher mitosis, and lower cyokeratin-5 expression while high nuclear and low cytoplasmic EHD2 signals showed a reverse correlation and was associated with ER/PR/AR-positive and non-TNBC status (*Supplementary file 1*-Table 1B). High cytoplasmic EHD2 predicted shorter BC-specific survival, while high nuclear EHD2 showed an opposite correlation (*Figure 3B*). Across BC subtypes, the high cytoplasmic and nuclear-negative EHD2, which also predicted shorter BC-specific survival (*Figure 3—figure supplement 1*), was seen in about half of TNBC and HER2 + samples, and a third of ER + samples (*Figure 3C*, *Supplementary file 1*-Table 1B). Analysis of a subset of our patient cohort with data for further subtyping showed a strong skewing of high cytoplasmic EHD2 expression in basal-like TNBC and to some extent the HER2-enriched and luminal B subtypes, while high nuclear EHD2 expression was a feature of luminal A subtype (*Figure 3D–E*). Thus, our results indicate that high cytoplasmic EHD2 expression, a localization similar to that observed in normal mouse mammary epithelium and human BC cell lines, is a marker of more aggressive BC, contrary to published reports that did not assess the cytoplasmic/nuclear distribution of EHD2 and suggested its potential tumor suppressor role (*Yang et al., 2015*; *Shen et al., 2020*; *Shi et al., 2015*).

## EHD2 knockdown or knockout in TNBC cell lines impairs the tumorigenic and pro-metastatic traits

To examine the role of EHD2 expression in BC oncogenesis, we established control or *EHD2* shRNA expressing TNBC cell lines, Hs578T, BT549 and MDA-MB-231 (*Figure 4A*). While EHD2 knockdown (KD) did not affect proliferation in two-dimensional culture on plastic (*Figure 4B*), it markedly reduced the tumorsphere growth under low attachment (*Figure 4C*), impaired invasion across Matrigel in trans-well assays (*Figure 4D*) and markedly reduced the invasive fronts in a Matrigel organoid invasion assay (*Figure 4E*).

Orthotopically implanted control Hs578T cells produced xenograft tumors over time while *EHD2* KD cells showed a severe reduction in tumor formation (*Figure 4F*). Immunostaining confirmed the *EHD2* KD (*Figure 4G*) and showed marked reduction in proliferation (Ki67[+]) with sparse tumor cells in H&E sections (*Figure 4H*). *EHD2* KD in MDA-MB-231 cells also reduced the xenograft growth and frequency of lung tumor metastasis (*Figure 4—figure supplement 1*).

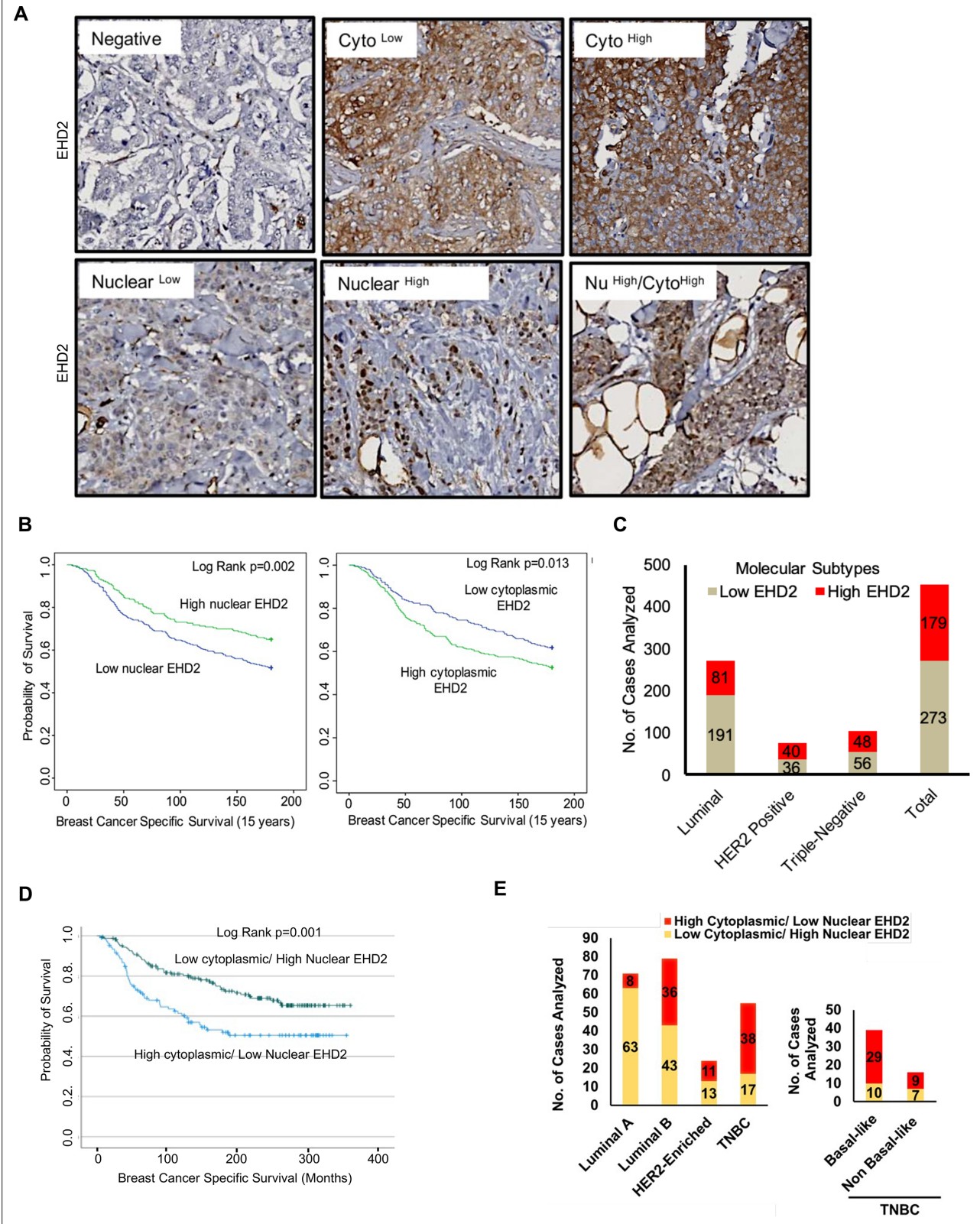

**Figure 3.** EHD2 is overexpressed in a subset of breast cancer patients and is associated with metastasis and shorter survival. (**A**) Representative images of negative/low/high cytoplasmic and nuclear EHD2 IHC staining of a breast cancer tumor microarray (840 samples). Magnification, 20X. (**B**) Kaplan-Meier survival curves correlating positive/high (green) vs. low/negative (blue) nuclear (left panel; N=288 vs 458) or cytoplasmic (right panel; N=392 vs 352) EHD2 expression with Breast Cancer Specific Survival (BCSS). (**C**) Number (Y-axis) of cytoplasmic EHD2-negative/low (gray) and -positive/

*Figure 3 continued on next page*

*Figure 3 continued*

high samples among ER/PR+, ErbB2+, TN, and all tumors. (**D**) Kaplan-Meier survival analysis of a subset of patients with molecular subtyping markers available (N=271) comparing high cytoplasmic/low nuclear (blue; N=107 out of 271) vs low cytoplasmic/high nuclear (green; N=164 out of 271). (**E**) Left panel - number (Y-axis) of high cytoplasmic/low nuclear EHD2 (red) and low cytoplasmic /high nuclear EHD2 (yellow) cases (among the 271 cases analyzed in D) within the luminal A (ER⁺/PR⁺, HER2⁻ and Ki67 <14%), luminal B (ER⁺/PR⁺ or ⁻ and either HER2⁺ or Ki67 >/=14% or both), HER2-Enriched (ER⁻, PR⁻ and HER2⁺, regardless of the Ki67) and TNBC (ER, PR and HER2⁻, regardless of the Ki67) BC subtypes. Right panel - number of high cytoplasmic/low nuclear (red) or low cytoplasmic/high nuclear (yellow) EHD2 staining in basal-like (CK5/6 or CK14 or CK17 positive) and non-basal-like (CK5/6, CK14 or CK17 negative) TNBC subtypes.

The online version of this article includes the following figure supplement(s) for figure 3:

**Figure supplement 1.** Kaplan-Meier survival curve for Breast Cancer Specific Survival (BCSS) probability in all tumors scored for cytoplasmic and nuclear positive (purple), cytoplasmic and nuclear negative (blue), cytoplasmic positive and nuclear negative (green), cytoplasmic negative and nuclear positive (yellow) EHD2 expression.

Further, CRISPR-Cas9-mediated *EHD2* knockout (KO) in TNBC cell lines (*Figure 5A*), which unlike our observations in *Ehd2*-KO mouse tissues was not associated with any significant changes in EHD1/4 expression (*Figure 5—figure supplement 1*), significantly impaired their migration, invasion (*Figure 5B–C*, *Figure 5—figure supplement 2A–B*) and extracellular matrix (ECM) degradation ability (*Figure 5D*, *Figure 5—figure supplement 2C*), another pro-metastatic trait (*Cheung and Ewald, 2014*). Introduction of mouse *Ehd2* in MDA-MB-231 *EHD2*-KO cells, at levels lower than in control cells, significantly rescued the cell migration defect (*Figure 5E*), demonstrating specificity. Reciprocally, CRISPR activation of endogenous *EHD2* in EHD2-nonexpressing MDA-MB-468 TNBC cells (*Figure 5F*) increased cell migration compared to control cells (*Figure 5G*). When orthotopically implanted in nude mice, *EHD2*-KO MDA-MB-231 cells exhibited a marked and significant defect in tumor formation, with a significant rescue upon mouse *Ehd2* expression (*Figure 5H*).

To directly assess the role of EHD2 in metastasis, luciferase-expressing control and KO MDA-MB-231 cells were intravenously injected into nude mice. Luminescence bioimaging showed time-dependent lung metastatic growth of control cells but no growth (or a reduction in signals) with *EHD2*-KO cells (*Figure 6A–C*). These findings were confirmed by assessment of lung metastatic nodules at necropsy (*Figure 6D*). H&E and human CK18 staining confirmed the metastatic growths, and EHD2 expression pattern was confirmed by IHC (*Figure 6E*). Collectively, our analyses definitively demonstrate a positive role of EHD2 in tumorigenic and pro-metastatic behavior in TNBC.

## EHD2 and CAV1/2 are co-overexpressed in basal-like breast cancer and loss of EHD2 reduces the cell surface caveolae

EHD2 localizes to and is required for the stability of the cell surface caveolae (*Morén et al., 2012*; *Stoeber et al., 2012*; *Torrino et al., 2018*; *Senju et al., 2015*). The bc-GenExMiner analysis of 5277 BC samples *Jézéquel et al., 2013* demonstrated tight co-expression of *EHD2* with the structural components of caveolae, *CAV1* and *CAV2* in TNBC samples (*Figure 7A*, *Figure 7—figure supplement 1A*). By KM Plotter analysis, combined EHD2-, CAV1-, and CAV2-high basal (PAM50-based) but not all BC patients showed significantly shorter distal metastasis-free survival (*Figure 7B*, *Figure 7—figure supplement 1B*). Immunoblotting demonstrated concordant EHD2 and CAV1 expression in mammary epithelial and BC cell lines (*Figure 7C*). Immunofluorescence analysis using structured illumination microscopy (SIM) demonstrated a high degree of colocalization between EHD2 and CAV1 in TNBC cell lines (*Figure 7D*). Total internal reflection fluorescence (TIRF) microscopy analysis showed a significant reduction of cell surface associated CAV1-GFP puncta, representing cell surface caveolae, in *EHD2*-KO compared to control Hs578T cells (*Figure 7E*), consistent with the reported electron microcopy-based high cell surface caveolae density on Hs578T compared to a lower density on the EHD2-non-expressing MDA-MB-436 cells (*Torrino et al., 2018*). CRISPR KO of *CAV1* (*Figure 7F*) led to a significant impairment of cell migration like that with *EHD2*-KO (*Figure 7G*). These results support the conclusion that EHD2-dependent maintenance of cell surface caveolae is linked to its promotion of tumorigenic and pro-metastatic traits, although the potential role(s) of EHD2 and caveolae in other subcellular locations cannot be excluded at present.

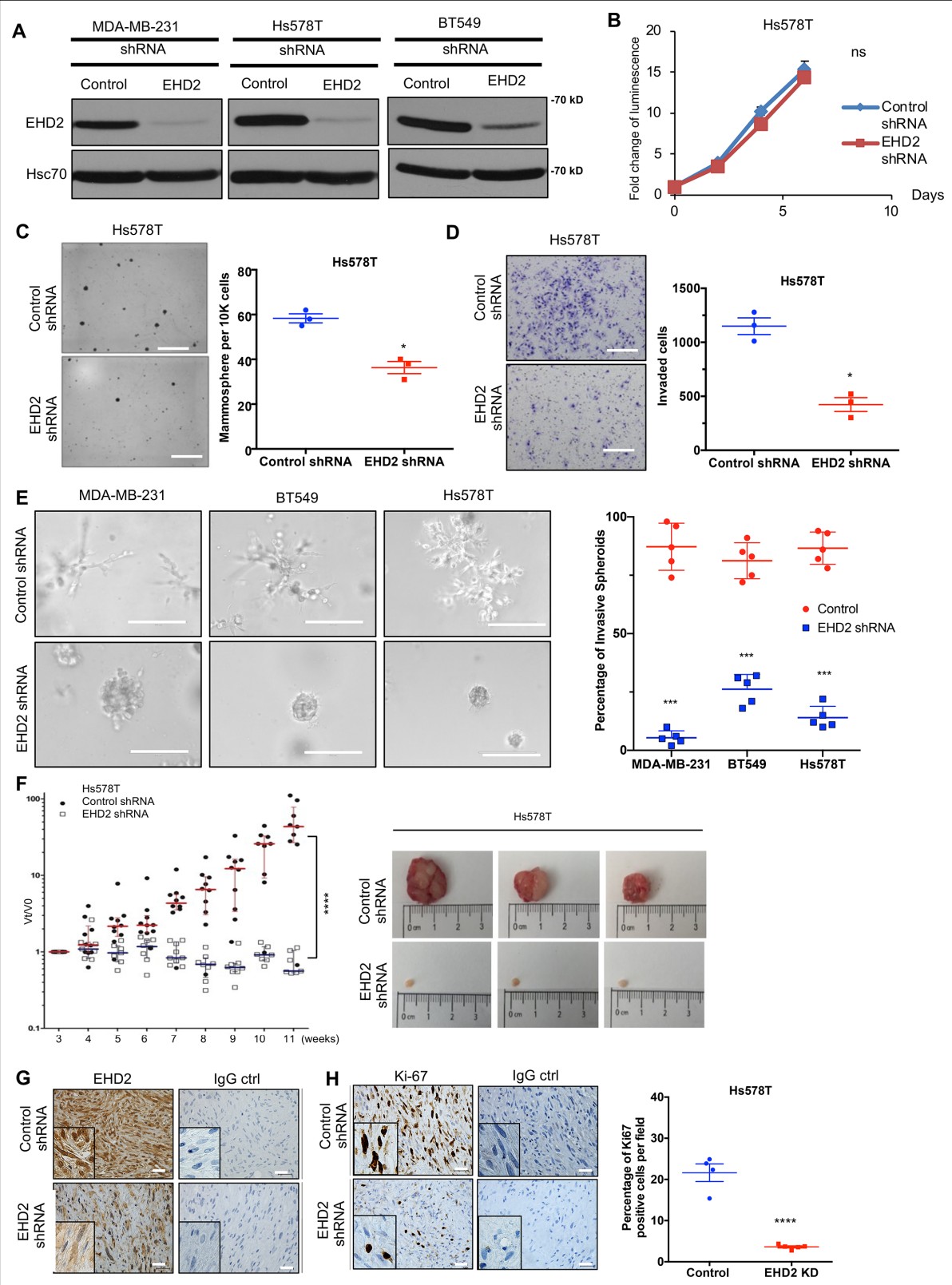

**Figure 4.** *EHD2* knockdown in TNBC cell lines impairs the tumorigenic and pro-metastatic traits. (**A**) Immunoblot confirmation of shRNA-mediated *EHD2* knockdown. (**B**) Cell Titer-Glo proliferation (2000 cells/well; 24 replicates each) over time. Mean +/- SEM, n=3, ns, not significant. (**C**) Tumorsphere formation quantified on day 7. Left, representative images; Right, quantification of tumorspheres/well. Mean +/- SEM, n=3, *p<0.05; **p<0.01. Scale bar, 400 μm. (**D**) Transwell invasion of cells plated in 0.5% FBS medium towards complete medium assayed after 18 hr. Left, representative images; Right,

*Figure 4 continued on next page*

*Figure 4 continued*

quantification of invaded cells (Mean +/- SEM, n=3, *p<0.05). Scale bar, 400 μm. (**E**) Three-dimensional invasion in Matrigel-grown organoids. A total of 2000 cells plated per well in 50% Matrigel on top of 100% Matrigel layer in eight-well chamber slides for 7 days before imaging. Left, representative images; right, % spheroids with invasive fronts from over 100 counted per well, n=4, *** p<0.001. Scale bar, 200 μm. (**F**) Xenograft tumorigenesis. Four-week-old nude mice orthotopically-injected with 5x10$^6$ cells were followed over time. Left, fold change in tumor volume over time for individual mice. Mean (red/blue lines) +/- SEM; ****p<0.0001 by two-way ANOVA. Right, representative tumors (close to median of groups). (**G, H**) Representative IHC staining of tumor sections for EHD2 (**G**) or Ki67 (**H**), with respective controls. Right, Mean +/- SEM of Ki67 + staining. ****, p<0.0001. Scale bar, 25 μm.

The online version of this article includes the following source data and figure supplement(s) for figure 4:

**Source data 1.** Original blots of *Figure 4A*.

**Figure supplement 1.** Knockdown of *EHD2* impaired the tumorigeneses and metastases in vivo.

**Figure supplement 1—source data 1.** Original table in *Figure 4—figure supplement 1*.

## EHD2 promotes pro-metastatic traits in TNBC cells by upregulating store-operated calcium entry

The impact of EHD2 depletion on multiple oncogenic traits and its regulation of plasma membrane caveolae suggested the role for a caveolae-linked signaling machinery. We investigated the linkage of EHD2 to store-operated calcium entry (SOCE) (*Chung et al., 2017*), a pathway that operates at caveolae (*Pani and Singh, 2009*; *Bohórquez-Hernández et al., 2017*) and is a well-established pro-metastatic signaling pathway in TNBC and other cancers (*Mo and Yang, 2018*; *Yang et al., 2009*). The SOCE is mediated by translocation of the endoplasmic reticulum (ER) Ca$^{2+}$ sensor stromal-interaction molecule 1 (STIM1) to ER-plasma membrane contact sites upon ER Ca$^{2+}$ depletion which permits its binding to and activation of the Orai1 membrane Ca$^{2+}$ channel to promote Ca$^{2+}$ entry for Ca$^{2+}$-dependent signaling and refilling of the ER (*Chung et al., 2017*).

To examine if EHD2 regulates SOCE in TNBC cells, calcium-sensitive fluorescent dye (Fluo4 AM)-loaded cells in Ca$^{2+}$-free medium were treated with thapsigargin (Tg), an inhibitor of the ER-localized Sarco-Endoplasmic Reticulum Ca$^{2+}$ ATPase 2 (SERCA-2) (*Peterková et al., 2020*). Expectedly, control Hs578T or BT549 TNBC cells exhibited a robust rise in cytoplasmic Ca$^{2+}$ that declined over time (*Figure 8A–B*), reflecting the release of ER Ca$^{2+}$ (*Ong et al., 2007*). Subsequent addition of Ca$^{2+}$ in the medium induced a rapid increase in cytoplasmic Ca$^{2+}$, indicating the SOCE (*Figure 8A–B*; *Ong et al., 2007*). Pre-treatment with the SOCE inhibitor SKF96365 (*Yang et al., 2009*) markedly reduced the initial Ca$^{2+}$ flux and nearly abrogated the SOCE (*Figure 8C*). *EHD2*-KO cells demonstrated a marked defect in both the initial Tg-induced rise in cytoplasmic Ca$^{2+}$ and the subsequent SOCE (*Figure 8A–B*), consistent with the established role of SOCE in intracellular Ca$^{2+}$ store filling (the source of the initial Ca$^{2+}$ release upon Tg treatment) besides Ca$^{2+}$-dependent signaling (*Chung et al., 2017*). Defective SOCE was also seen in *EHD2*-KO Hs578T cells using another SERCA inhibitor cyclopiazonic acid (CPA) (*Demaurex et al., 1992*; *Figure 8D*). In a genetic approach, we showed that *EHD2*-KO Hs578T cells stably expressing a GFP-based reporter of cytoplasmic Ca$^{2+}$, GCaMP6s, (*Chen et al., 2013*), exhibited defective Tg-induced SOCE (*Figure 8E*, *Figure 8—figure supplement 1A*). Further, stable expression of GCaMP6s-CAAX, a plasma membrane-targeted fluorescent reporter of Ca$^{2+}$ levels (*Tsai et al., 2014*), which only detects the SOCE phase upon Tg treatment directly established the defective SOCE in *EHD2*-KO Hs578T cells (*Figure 8F*, *Figure 8—figure supplement 1B*). In a reciprocal experiment, CRISPRa- induced endogenous *EHD2* expression in EHD2-negative MDA-MB-468 cells led to a marked increase in Tg-induced SOCE (*Figure 8G*). Consistent with the role of caveolae, a marked defect in SOCE was observed in *CAV1*-KO TNBC cell lines (*Figure 8H–I*).

Next, we transiently transfected the CFP-tagged *STIM1* in control or *EHD2*-KO Hs578T cells and quantified the number of fluorescent STIM1 puncta at the cell surface, a measure of STIM1-Orai1 interaction, using TIRF microscopy (*Wu et al., 2006*). Tg treatment failed to increase the STIM1 puncta in *EHD2*-KO cells (*Figure 9A*). This defect was not a result of reduced levels of total STIM1 and Orai1 proteins (*Figure 9B*). Given the known localization of Orai1 in caveolae (*Bohórquez-Hernández et al., 2017*; *Sathish et al., 2012*), we assessed the impact of *EHD2*-KO on Orai1 cell surface levels. We used an anti-Orai1 antibody authenticated against control or Orai1 knockdown TNBC cell lines (*Figure 9—figure supplement 1*) to immunoprecipitate Orai1 from surface biotin-labeled control and *EHD2*-KO MDA-MB-231 or Hs578T cells and confirmed the comparable immunoprecipitation of total Orai1 in WT vs. KO cells (*Figure 9C*, **lower panels**). In contrast, streptavidin blotting revealed

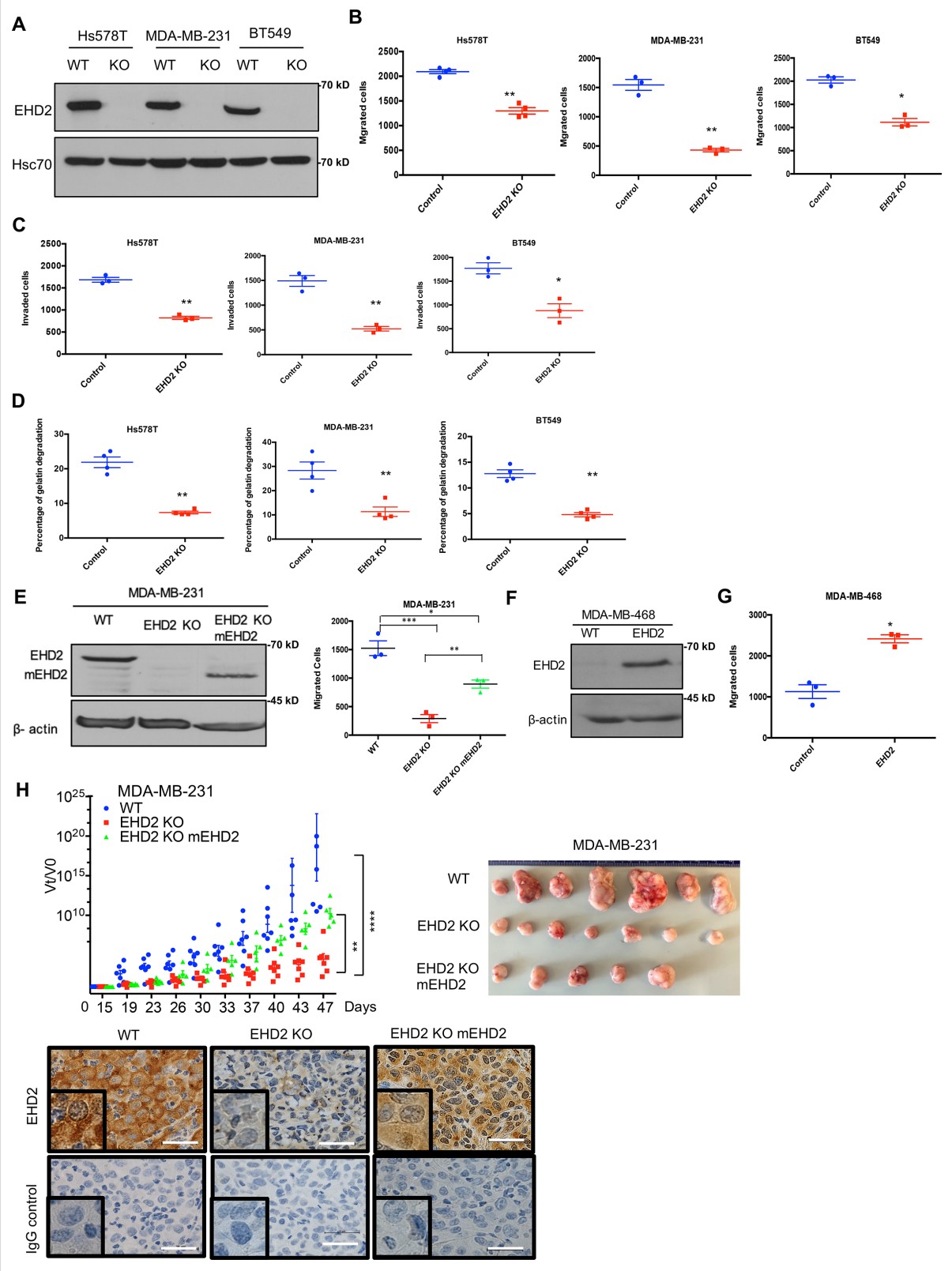

**Figure 5.** *EHD2* knockout in TNBC cell lines impairs the tumorigenic and pro-metastatic traits. Single cell clones of TNBC cell lines serially transduced with Cas9 and control or *EHD2* sgRNA lentiviruses were obtained and used as a pool of >3 clones. (**A**) Immunoblotting of EHD2 expression in KO vs. WT (Cas9) controls. (**B**) Transwell migration. Data points are independent experiments; Mean +/- SEM of migrated cells (input 10 K), **p<0.01, *p<0.05. (**C**) Transwell invasion across Matrigel. Mean +/- SEM of invaded cells (input 10 K), **p<0.01, *p<0.05. (**D**) Extracellular matrix degradation. Cells plated

*Figure 5 continued on next page*

*Figure 5 continued*

on Cy5-gelatin and percentage area with matrix degradation quantified after 48 hr. Mean +/- SEM, **p<0.01. (**E**) Mouse *Ehd2* rescue of *EHD2*- KO MDA-MB-231 cells. Left, immunoblot to show re-expression of mouse EHD2; beta-actin, loading control. Right, rescue of cell migration defect. Mean +/- SEM, ***p<0.001,   **p<0.01, *p<0.05. (**F–G**) CRISPRa induction of endogenous *EHD2* expression in EHD2-negative MDA-MB-468 cell line (**F**) and increase in migration (**G**). Mean +/- SEM, *p<0.05. (**H**) Impairment of tumorigenesis by *EHD2*-KO and rescue by mouse *Ehd2* reconstitution. Left, groups of eight nude mice orthotopically implanted with 3x10⁶ cells and tumors analyzed as in *Figure 4F*: ****p<0.0001, **p=0.001. Right, Representative tumor images. Bottom, representative tumor sections stained for EHD2 and control. Scale bar, 25 µm.

The online version of this article includes the following source data and figure supplement(s) for figure 5:

**Source data 1.** Original blots of *Figure 5A*.

**Source data 2.** Original blots of *Figure 5E*.

**Source data 3.** Original blots of *Figure 5F*.

**Figure supplement 1.** EHD1/4 expression is unchanged in *EHD2* knockout TNBC cell lines.

**Figure supplement 1—source data 1.** Original blots of *Figure 5—figure supplement 1A*.

**Figure supplement 2.** Loss of EHD2 decreased oncogenesis traits in TNBC cells.

a marked reduction in biotinylated (cell surface) Orai1 signals in *EHD2*-KO cells (*Figure 9C*, **upper panels**). Further linking the SOCE to EHD2-dependent pro-metastatic traits, overexpression of CFP-STIM1 in *EHD2*-KO Hs578T cells (*Figure 9D*) partially rescued the SOCE defect (*Figure 9E*) and the defective cell migration (*Figure 9F*). In a complementary approach, the tool SOCE inhibitor SKF96365 and a recently identified inhibitor CM4620, which (as Auxora, CalciMedica) has progressed to phase 3 clinical trials in acute inflammatory disease conditions (*Bruen et al., 2022*), significantly impaired the wildtype TNBC cell migration, and further reduced the migration of *EHD2*-KO cells, albeit the latter was not statistically significant (*Figure 10A*). Thus, a major proportion of the SOCE in TNBC cell lines is dependent on EHD2 and is inhibitable with available SOCE inhibitors. Accordingly, we show that SKF96365 treatment significantly reduced the control TNBC xenograft tumor growth (*Figure 10B*); we were unable to test the SOCE inhibitor against *EHD2*-KO TNBC xenografts as these did not grow sufficiently to test the inhibitor impact. Collectively, these findings support our conclusion that EHD2, by stabilizing caveolae, facilitates the SOCE to promote downstream pro-oncogenic traits in TNBC cells.

## Discussion

Elucidating novel tumorigenesis- and metastasis-associated cellular adaptations could dictate new therapeutic options in BC. Here, we use TNBC cell models to elucidate a novel signaling axis linking EHD2 overexpression in BC to store-operated calcium entry (SOCE), a known pro-oncogenic and pro-metastatic pathway. Our studies support the potential for targeting the SOCE pathway in EHD2-overexpressing TNBC and other BC subtypes.

Our IHC analyses demonstrated high cytoplasmic EHD2 expression in a substantial proportion of breast tumors, associated with shorter BC-specific patient survival (*Figure 3B*) and higher tumor grade (*Supplementary file 1*-Table 1). A higher proportion of TNBC, luminal B and HER2 + patients exhibited high cytoplasmic EHD2 (*Figure 3D*, *Supplementary file 1*-Table 1). Our results differ from reports that reduction in EHD2 expression correlates with tumor progression in BC (*Yang et al., 2015*; *Shi et al., 2015*). Notably, a recent study, while it reported the depletion of EHD2 to increase the oncogenic traits of BC cell lines in vitro, found low EHD2 expression in breast tumors to specify good prognosis and better chemotherapy response (*Shen et al., 2020*). Several factors could account for the discordance, including the lack of validation of antibodies in prior studies, the high EHD2 expression in normal mammary adipocytes (*Figure 1B*), resulting in apparent reduction in EHD2 expression in tumor tissue using western blotting (*Yang et al., 2015*; *Shen et al., 2020*; *Shi et al., 2015*), and the possibility that EHD2 signals in prior studies represented nuclear EHD2, which we find is associated with positive prognostic factors (*Figure 3B*, *Supplementary file 1*-Table 1). Previous cell-line-based studies have shown that EHD2 shuttles between the nucleus and cytoplasm based on nuclear import and exit signals, the latter dependent on its SUMOylation, and that nuclear EHD2 repressed transcription from fused transactivation domains (*Pekar et al., 2012*). Another study found that application of mechanical stress to cells led to EHD2 release from caveolae, its SUMOylation and nuclear

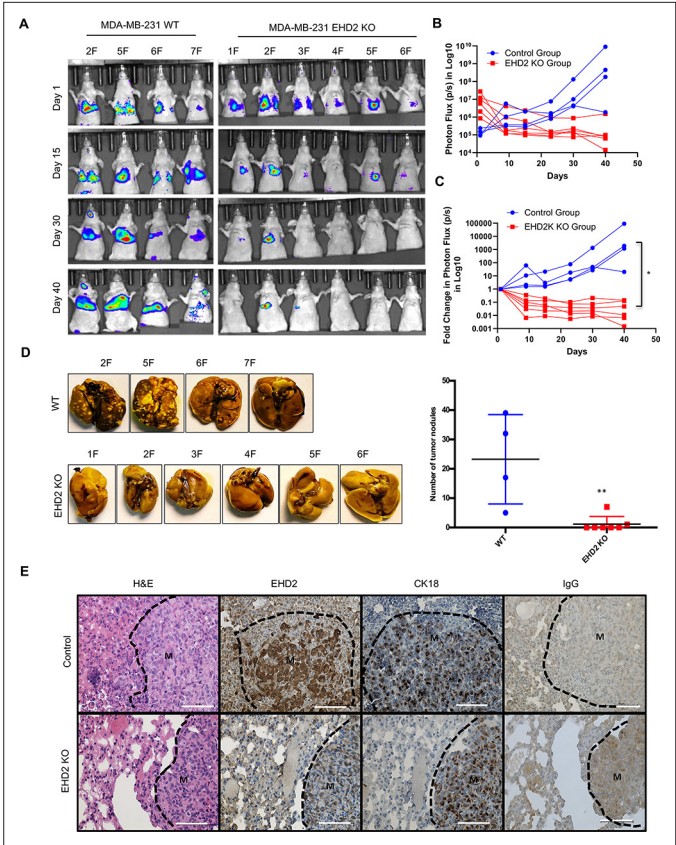

**Figure 6.** *EHD2* KO impairs the ability of TNBC cells to form lung metastases. WT control and *EHD2*-KO MDA-MB-231 cells were engineered with tdTomato-luciferase and 10⁶ cells of each injected intravenously into groups of seven nude mice. Lung metastases were monitored by bioluminescence imaging (**A**) Bioluminescence images of mice over time. . (**B–C**) Bioluminescence signals over time (Control, blue; KO, red) are shown as either untransformed photon flux values (**B**) or log fold-change in photon flux relative to day 0 (**C**). Two-way ANOVA showed the differences between Control and KO groups to be significant (*$p<0.05$). (**D**) Left panel, images of lungs harvested at necropsy show nearly complete absence of metastatic nodules in lungs of mice injected with *EHD2*-KO cells. Right panel, quantification of tumor nodules in the lungs, **, $p<0.01$. (**E**) Representative H&E (first panels), EHD2 (second panels), CK18 (third panels) and control IgG staining (fourth panels) of metastatic lung tissue sections from control (upper) and *EHD2*-KO cell injected mice. Note the retention of normal lung tissue in *EHD2*-KO cell injected mouse lung, and absence of EHD2 expression in KO nodules (labeled M). CK18 demarcates the human tumor cell area. Scale bar, 50 μm.

translocation where it regulated transcription of several genes including caveolar components, in a potential positive feedback mechanism (*Torrino et al., 2018*). However, no roles have thus far been assigned to the nuclear/cytoplasmic partitioning of EHD2 in oncogenesis. In view of the opposite prognostic significance of nuclear vs. non-nuclear EHD2 in our analyses of BC patient samples, it could be speculated that nuclear translocation may sequester EHD2 to inhibit its plasma membrane-associated pro-oncogenic role in SOCE, but rigorous studies will be needed to test this suggestion.

Analysis of mRNA expression in public BC databases (*Figure 7A*) and of protein levels in BC cell lines (*Figure 7C*) demonstrated high degree of EHD2 co-expression with caveolin-1/2, the structural elements of caveolae. This was noteworthy since EHD2 is known to regulate the stability of caveolae (*Morén et al., 2012*; *Stoeber et al., 2012*). Significantly, *EHD2* and *CAV1* or *CAV2* mRNA overexpression predicted shorter patient survival specifically in the PAM50-defined basal BC (*Figure 7B*), consistent with the predominant basal (myoepithelial) cell expression of EHD2 in mouse mammary epithelium (*Figure 1B and D*). Additional analyses of publicly available cell line mRNA data in CCLE (*Figure 2C*) and of a subset of BC samples analyzed for EHD2 expression (*Figure 2A*), further reinforce the conclusion that high cytoplasmic EHD2 expression is a feature of BC with basal-like/triple-negative features.

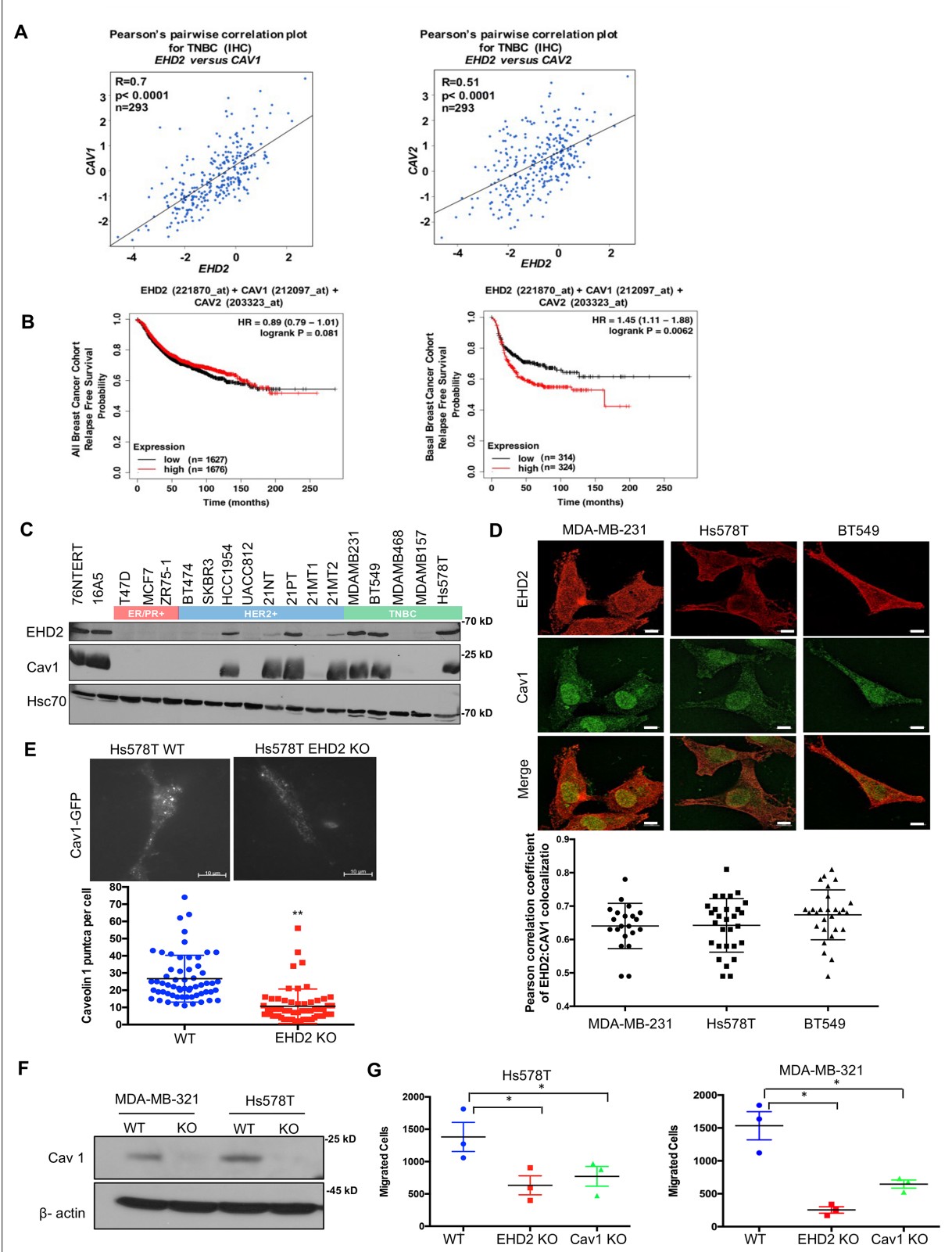

**Figure 7.** EHD2 and Caveolin-1/2 are co-overexpressed in breast cancers and EHD2 regulates cell surface caveolae. (**A**) Pearson's correlation plots of EHD2/CAV1 and EHD2/CAV2 expression in TNBC (IHC-based) subsets of TCGA and SCAN-B RNAseq datasets analyzed on bc-GenExMiner v4.5 platform. Indicated: n, number of samples; R, correlation coefficients; significance. (**B**) KM plotter analysis of EHD2, CAV1 and CAV2 overexpression correlation with relapse-free survival (RFS) for upper vs. lower quartiles in basal-like breast cancer (PAM50-based) cohorts of TCGA, GEO, and GEA

*Figure 7 continued on next page*

*Figure 7 continued*

datasets. Probe sets used: *EHD2* (221870_at), *CAV1* (212097_at) and *CAV2* (203323_at). Analysis of all samples combined found no survival differences (left panel). (**C**) Immunoblot analysis of coordinate EHD2 and CAV1 expression in immortal mammary epithelial cells and breast cancer cell lines. (**D**) SIM images demonstrated colocalization of EHD2 (red) and caveolin-1 (green) in TNBC cell lines; scale bar, 10 µm. Top, representative SIM images; Bottom, Pearson's Coefficient of Colocalization between EHD2 and CAV1 in TNBC cells from three independent experiments. (**E**) TIRF analysis of fluorescent CAV1 puncta to quantify cell surface caveolae pool. Top, representative TIRF images. Bottom, quantification of CAV1 puncta. Mean +/- SEM of puncta per cell pooled from 3 independent experiments; **p<0.01. Scale bar, 10µm. (**F**) Immunoblot confirmation of CRISPR-Cas9 *CAV1*-KO in TNBC cell lines. (**G**) Impact of *CAV1*-KO on Transwell migration. Mean +/- SEM number of migrated cells (input 10 K) per Transwell (n=3, *p<0.05).

The online version of this article includes the following source data and figure supplement(s) for figure 7:

**Source data 1.** Original blots of *Figure 7C*.

**Source data 2.** Original blots of *Figure 7F*.

**Figure supplement 1.** EHD2 and Caveolin-1/2 are correlated in breast cancers patients.

Multi-pronged approaches using shRNA knockdown and CRISPR-Cas9 KO of *EHD2* in TNBC cell models together with mouse *Ehd2* rescue of *EHD2*-KO demonstrated that EHD2 is required for tumorigenesis and metastasis. We show that in vitro tumor cell growth under stringent conditions (tumorspheres; *Figure 4C*) and pro-metastatic traits of cell migration, invasion, and ECM degradation (*Figures 4D–E, 5B–D*) are EHD2-dependent. In vivo, loss of EHD2 markedly impaired the orthotopic TNBC xenograft formation and metastasis (*Figure 4F*; *Figure 5H*), and tumor growth was rescued by exogenous mouse *Ehd2* (*Figure 5H*). Notably, intravenous injections demonstrated the inability of *EHD2*-KO TNBC cells to form lung metastases (*Figure 6*). Although further studies are needed to fully dissect the steps at which EHD2 is critical in the metastatic process, collectively, our analyses conclusively demonstrate that extranuclear EHD2 overexpression in BC cells represents a key pro-tumorigenic and pro-metastatic adaptation.

Mechanistically, we link the EHD2 overexpression in BC cells to regulation of caveolae, whose stability is known to be controlled by EHD2 (*Morén et al., 2012*; *Stoeber et al., 2012*; *Morén et al., 2019*). This includes the strong EHD2 co-localization with CAV1/2 (*Figure 7D*), reduction in cell surface caveolae density using TIRF microscopy upon *EHD2*-KO (*Figure 7E*), and inhibition of TNBC cell migration upon *CAV1*-KO (*Figure 7G*), consistent with the previously documented pro-tumorigenic roles of CAV1 in TNBC (*Badana et al., 2016*; *Zou et al., 2017*). However, since our localization analyses show both EHD2 and CAV1 to also localize intracellularly besides at the plasma membrane (*Figure 7D*), and prior studies have identified pro-oncogenic roles for CAV1 localized in various intracellular locations (*Simón et al., 2020*), further studies will be needed to determine if the role of EHD2 we define here is exclusively related to regulation of the plasma membrane caveolae or might involve CAV1 in other compartments as well.

Caveolae serve as hubs for signaling (*Lamaze et al., 2017*). Among these, the SOCE pathway stood out as it is known to regulate multiple tumorigenic and pro-metastatic traits in TNBC (*Pani and Singh, 2009*; *Bohórquez-Hernández et al., 2017*), as with EHD2 depletion. Also, EHD2 interacts with $Ca^{2+}$-binding proteins such as Ferlins (*Posey et al., 2011*) that are involved in $Ca^{2+}$-dependent membrane repair and EHD2 was found to accumulate at sites of membrane repair in skeletal muscle models (*Marg et al., 2012*; *Demonbreun et al., 2016*). Indeed, our extensive analyses demonstrate that EHD2 is a major positive regulator of SOCE in TNBC cell models. This includes analyses of fluorescent dye-labeled cells and two distinct agents (thapsigargin or CPA) to release ER $Ca^{2+}$ as stimuli (*Figure 8A–B*), and independent validation using cytoplasm- or plasma membrane-localized genetic reporters of $Ca^{2+}$ (*Figure 8E–F*, *Figure 8—figure supplement 1A–B*). Reciprocally, CRISPR-activation of endogenous *EHD2* expression in an EHD2-nonexpressing TNBC cell line upregulated SOCE (*Figure 8G*) and cell migration (*Figure 5G*). Complementing these, *EHD2*-KO reduces the STIM1-Orai1 interaction at the ER-plasma membrane contact sites as measured using fluorescent STIM1 (*Figure 9A*) and overexpression of STIM1 partially rescues the SOCE and cell migration defects in *EHD2*-KO TNBC (*Figure 9E–F*). Consistent with the established role of SOCE in the intracellular $Ca^{2+}$ store refilling (*Chung et al., 2017*), which then serves as the source of the initial $Ca^{2+}$ release upon appropriate stimuli (Tg or CPA in our studies), loss of EHD2 or CAV1 expression or treatment with SOCE inhibitors led to an impairment of the SOCE as well as the initial $Ca^{2+}$ peak (*Figure 8A–E*; *Figure 8H–I*). Reciprocally, CRISPRa upregulation of *EHD2* elevated both phases of $Ca^{2+}$ flux (*Figure 8G*).

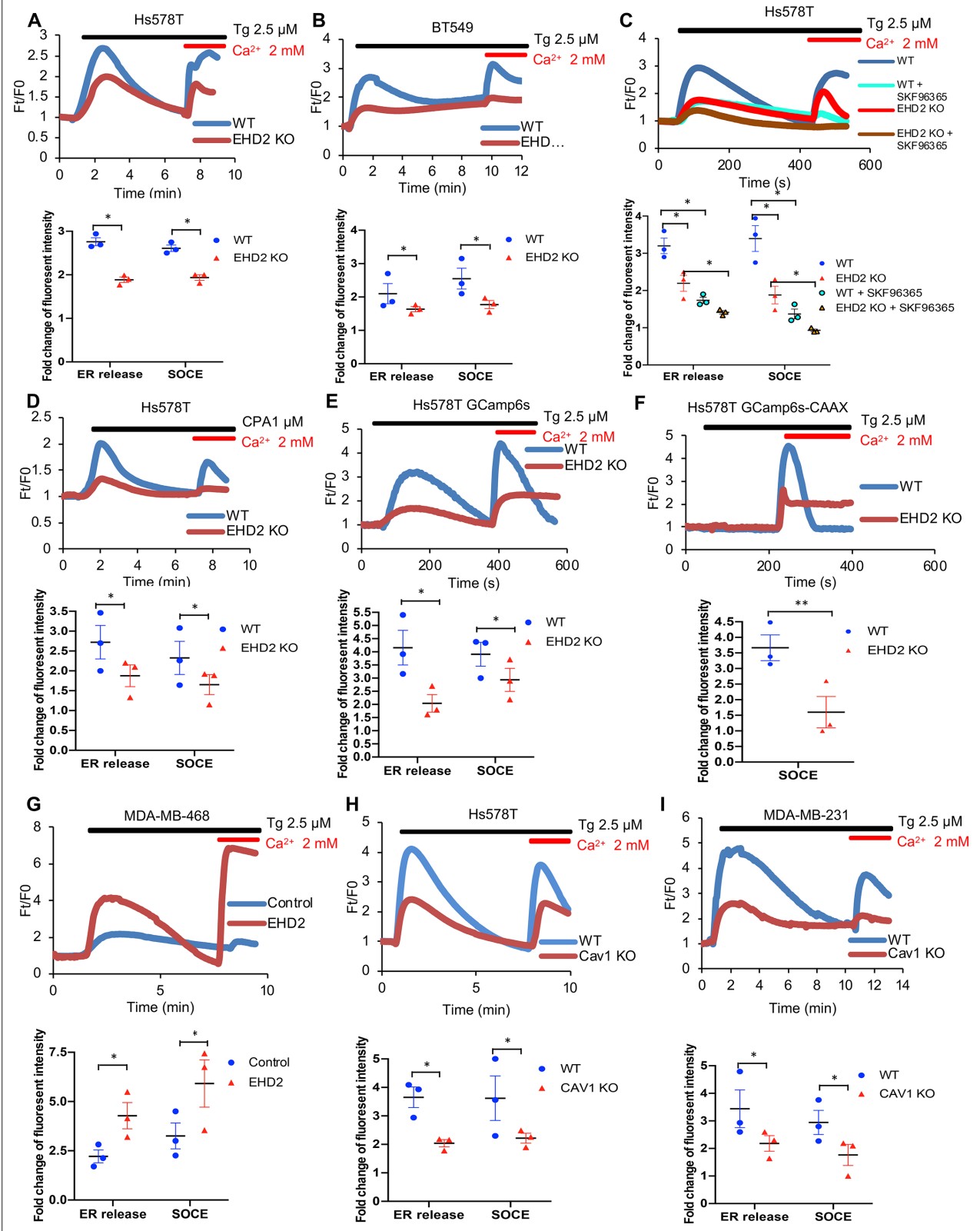

**Figure 8.** EHD2 promotes store-operated calcium entry (SOCE) in TNBC cell lines. (**A–B**) Thapsigargin (Tg; 2.5 μM)-induced increase in cytoplasmic $Ca^{2+}$ (initial rise in no extracellular $Ca^{2+}$) and SOCE (second peak after adding 2 mM $Ca^{2+}$) in Fluo 4 AM-loaded WT/KO Hs578T (**A**) or BT549 (**B**) cell lines measured by live-cell confocal microscopy. (**C**) Impact of SOCE inhibitor SKF96365 (10 μM) on Tg (2.5 μM)-induced $Ca^{2+}$ fluxes measured as in A. (**D**) Defective Tg-induced $Ca^{2+}$ fluxes demonstrated using cyclopiazonic acid (CPA; 1 μM). (**E–F**) Tg (2.5 μM)-induced $Ca^{2+}$ fluxes measured by confocal

*Figure 8 continued on next page*

*Figure 8 continued*

imaging of stably expressed genetic cytoplasmic Ca$^{2+}$ sensors: cytoplasmic sensor GCaMP6s (**E**) and plasma membrane-localized sensor GCaMP6s-CAAX (**F**). (**G**) Tg (2.5 μM)-induced Ca$^{2+}$ fluxes in Fluo4 AM-loaded control MDA-MB-468 (EHD2-negative) vs its CRISPRa derivative (EHD2-expressing). (**H–I**) Tg (2.5 μM)-induced Ca$^{2+}$ fluxes in Fluo4 AM-loaded control and *CAV1*-KO TNBC lines. Mean +/- SEM of peak fluorescence intensity (n=3, *p<0.05) is shown below all panels.

The online version of this article includes the following figure supplement(s) for figure 8:

**Figure supplement 1.** Loss of EHD2 decreased SOCE in TNBC cells.

Orai1 is a major STIM1-interacting caveolae-resident SOCE channel (*Bohórquez-Hernández et al., 2017*; *Sathish et al., 2012*). Indeed, our cell surface biotinylation studies demonstrated that *EHD2*-KO specifically reduces the cell surface Orai1 levels (*Figure 9C*). Thus, our findings support a model whereby EHD2-dependent stabilization of cell surface caveolae ensures high cell surface levels of Orai1 to enable robust SOCE in TNBCs, which in turn promotes pro-tumorigenic and pro-metastatic behaviors of tumor cells (*Mo and Yang, 2018*; *Yang et al., 2009*). Consistent with this model, EHD2 deficiency reduced the cell surface levels of caveolae-associated ATP-sensitive K$^+$ channels (*Yang et al., 2018*). However, further genetic studies to perturb Orai1 and its family members as well as other potential mediators of SOCE will be necessary to fully establish a causal role of the SOCE pathway in EHD2-dependent oncogenesis.

Finally, consistent with prior studies (*Yang et al., 2009*), chemical inhibition of SOCE markedly impaired the pro-metastatic traits of EHD2-overexpressing TNBCs, with a smaller impact on *EHD2*-KO cell lines (*Figure 10A*) and impaired the TNBC metastatic growth in vivo (*Figure 10B*). However, since SKF96365 targets additional Ca$^{2+}$ channels besides Orai1 (*Ramsey et al., 2006*; *Ding et al., 2012*), further studies using more selective inhibitors together with genetic analyses will be needed.

Together, our studies support the idea that EHD2-overexpressing subsets of TNBC and other BC subtypes may be selectively amenable to SOCE targeting, with EHD2 and CAV1/2 overexpression as predictors of response.

## Materials and methods

### Cell lines and medium

All breast cancer cell lines were obtained from ATCC and cultured in complete α-MEM medium with 5% fetal bovine serum, 10 mM HEPES, 1 mM each of sodium pyruvate, nonessential amino acids, and L-glutamine, 50 μM 2-ME, and 1% penicillin/ streptomycin (Life Technologies, Carlsbad, CA). The TNBC cell lines BT549 and Hs578T were cultured in α-MEM medium supplemented as above and with 1 μg/mL hydrocortisone and 12.5 ng/mL epidermal growth factor (Millipore Sigma, St. Louis, MO). The hTERT-immortalized 76NTERT (*Zhao et al., 2010*) and spontaneously immortalized MCF10A human mammary epithelial cell lines were maintained in DFCI-1 medium (*Band et al., 1990*), which contains 12.5 ng/ml EGF. Cell lines were maintained for less than 90 days in continuous culture and were regularly tested for mycoplasma.

### Antibodies and reagents

Antibodies used for immunoblotting were as follows: HSC70 (# sc-7298) and TRPC1 (# sc-133076) from Santa Cruz Biotechnology; STIM1 (# ab57834) from Abcam; Orai1 (# O8264) and beta-actin (# SAB1305567) from Millipore-Sigma; Caveolain-1 (#610057) from BD Biosciences. In-house generated Protein G-purified rabbit polyclonal rabbit anti-EHD2 antisera has been described previousl (*George et al., 2007*). The horseradish peroxidase (HRP)-conjugated Protein A or HRP-conjugated rabbit anti-mouse secondary antibody for immunoblotting were from Invitrogen. The alpha smooth muscle actin (# ab7817), cytokeratin 8 (#53280), cytokeratin 18 (# 133263), Ki67 (# ab16667) antibodies for immunohistochemistry (IHC) and immunofluorescence (IF) staining were from Abcam. Thapsigargin (# T7459) and Fluo 4AM (# 14201) were from Thermo Fisher Scientific. Cyclopiazonic acid (# C1530) and D-luciferin (#L9504) were from Millipore Sigma. SKF96365 (cat. # S7999), SOCE inhibitor CM4620 (# S6834) from SelleckChem, Matrigel (# 356230) from Corning, and Isoflurane (# 502017) from MWI Animal Health.

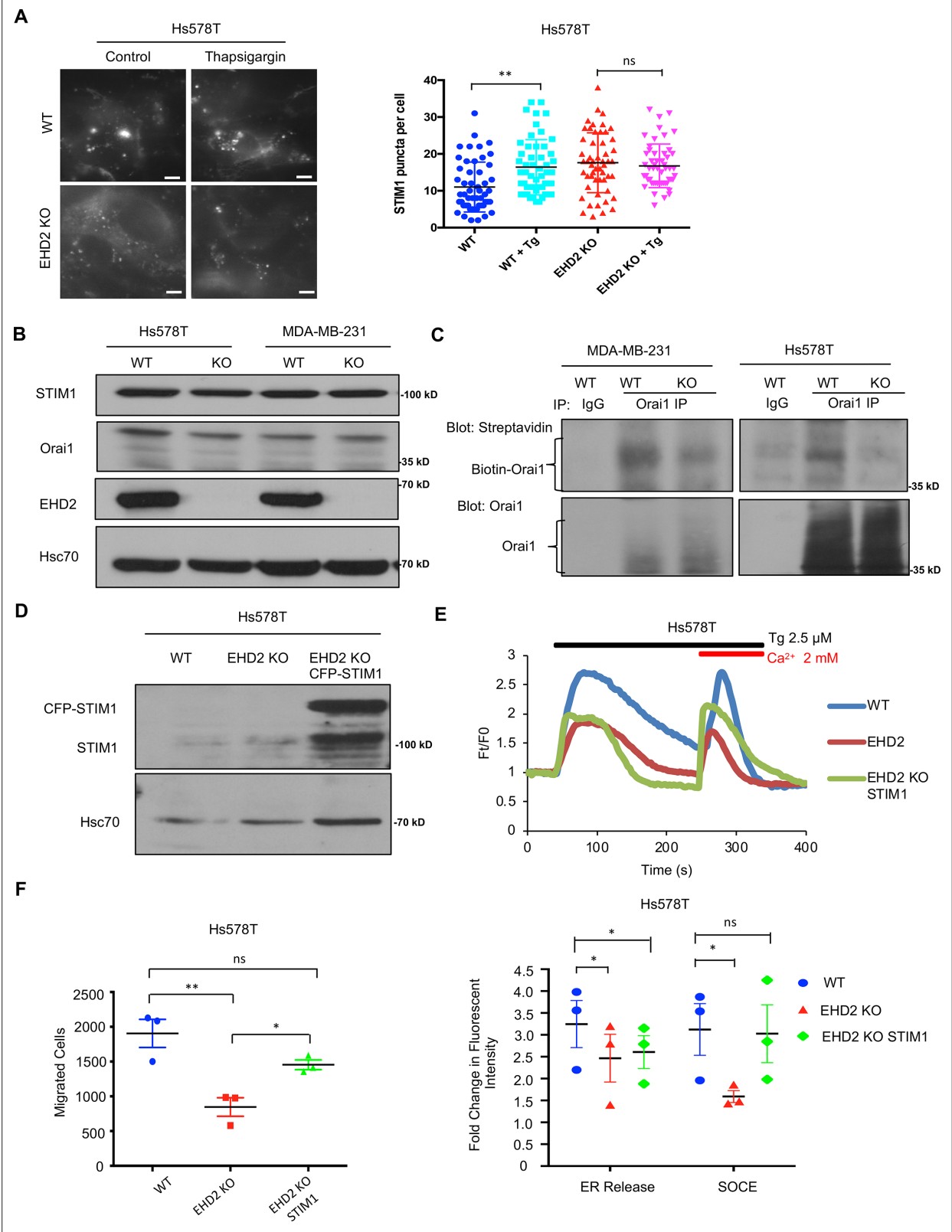

**Figure 9.** EHD2 regulates SOCE through STIM1-Orai1. (**A**) CFP-STIM1-trasnfected cells were analyzed for plasma membrane proximal fluorescent puncta by TIRF microscopy, without (control) or with thapsigargin treatment (2.5 µM, 5 min). Left, representative TIRF images; Right, Mean +/- SEM of STIM1 puncta/cell, ** p<0.01. Scale bar, 5 µm. (**B**) Immunoblotting to show comparable total STIM1 and Orai1 levels in control vs *EHD2*-KO TNBC lines; Hsc70, loading control. (**C**) Reduced cell surface levels of Orai1 in *EHD2*-KO cells. Live cell surface biotinylated cell Orai-1 immunoprecipitates blotted

*Figure 9 continued on next page*

*Figure 9 continued*

with Streptavidin (top) and Orai1 (bottom). (**D**) Anti-STIM1 immunoblotting to show stable overexpression of STIM1-CFP in *EHD2*-KO Hs578T cells. (**E**) Partial rescue of SOCE by ectopic CFP-STIM1 overexpression analyzed upon thapsigargin (Tg; 2.5 µM) treatment of Fluo 4 AM-loaded cells. Bottom, Mean +/- SEM of peak fluorescence, N=3; *p<0.05. (**F**) Partial rescue of Transwell cell migration defect by CFP-STIM1 overexpression in *EHD2*-KO cells. Mean +/- SEM of migrated cells (input 10 K); n=3; *p<0.05.

The online version of this article includes the following source data and figure supplement(s) for figure 9:

**Source data 1.** Original blots of *Figure 9B*.

**Source data 2.** Original blots of *Figure 9C*.

**Source data 3.** Original blots of *Figure 9D*.

**Figure supplement 1.** Validation of ORAI1 antibody in triple negative breast cancer cell lines.

**Figure supplement 1—source data 1.** Original blots of *Figure 9—figure supplement 1A*.

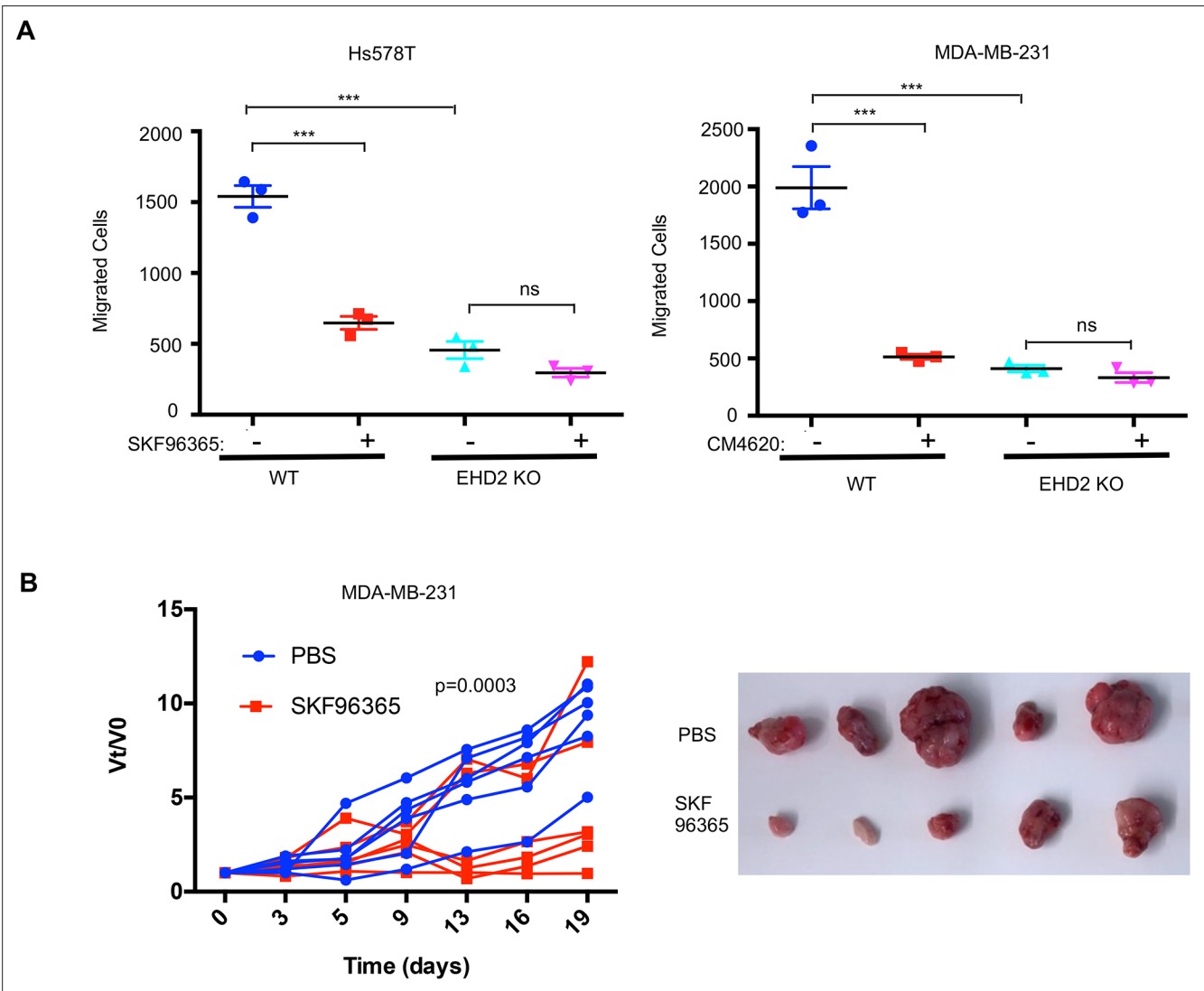

**Figure 10.** EHD2 expression determines the relative functional impact of SOCE inhibition in TNBC cells. (**A**) Impact of SOCE inhibitors SKF96365 (10 µM) or CM4620 (10 µM) Transwell migration of Control vs *EHD2*-KO TNBC cell lines. Mean +/- SEM of n=3; ***p<0.001. (**B**) SOCE inhibition reduces TNBC tumorigenesis. Nude mice (8/group) bearing orthotopic MDA-MB-231 (3x10^6) cell tumors (average 4–5 mm in diameter) were administered 10 mg/kg SKF96365 (in PBS) or PBS intraperitoneally and change in tumor volumes ($V_t/V_0$) monitored over time and differences analyzed by two-way ANOVA. Right, representative tumor images.

## Transfection reagents and plasmids

XtremeGENE 9 transfection reagent was from Roche Applied Science (Indianapolis, IN); Commercial pQCXIX-RT3GEP (*Fellmann et al., 2013*) shRNA construct for scrambled shRNA or *EHD2*-targeted shRNA (Sequence 1, gaaggctcgagaaggtatattgctgttgacagtgagcgCTCACGCTTACATCATCAGCTATA GTGAAGCCACAGATGTATAGCTGATGATGTAAGCGTGAAtgcctactgcctcggacttcaaggggctagaattcga gca; Sequence 2, gaaggctcgagaaggtatattgctgttgacagtgagcgCTCCATCCGTCATTCATTCAAATAGT GAAGCCACAGATGTATTTGAATGAATGACGGATGGATtgcctactgcctcggacttcaaggggctagaattcgagc a; uppercase letters indicate the target sequence) were custom-made through Mirimus (Brooklyn, NY). Human STIM1-CFP plasmid (*Prakriya et al., 2006*) was from Addgene (#19755). CAV1-mEGFP plasmid (*Hayer et al., 2010*) was from Addgene (#27704). Lentiviral Mouse *Ehd2* vector (pLenti-GIII-CMV-GFP-2A-puro, cat. # 190520640395) was from Applied Biological Materials (Richmond, BC, Canada). Luciferase/tdTomatao reporter was engineered using the MuLE system kit from Addgene (Cat. # 1000000060) (*Albers et al., 2015*).

## Generation of shRNA knockdown cell lines

Retroviral production and infection was carried out as described previously (*Nadeau et al., 2017*). Briefly, for retroviral expression of control and *EHD2* shRNA, HEK-293T cells were transfected with pQCXIX-RT3GEP vectors harboring control shRNA or *EHD2* shRNAs together with packaging vectors (plK was a gift from Dr. David Root, Broad Institute), and supernatants used to infect triple negative breast cancer cell lines followed by selection in puromycin.

## Generation of CRISPR-Cas9 knockout/activation cell lines

Edit-R Lentiviral Cas9 nuclease and Edit-R Lentiviral sgRNA control and *EHD2* vectors (# SO-2646983G, Dharmacon) were used in a two-step transduction process to derive CRISPR-Cas9 *EHD2*-KO cell lines. CAV1 sgRNA CRISPR/Cas9 All-in-One Lentivector (pLenti-U6-sgRNA-SFFV-Cas9-2A-Puro) from Applied Biological Materials (Richmond, BC, Canada) was used to derive *CAV1*-KO cell lines. For induction of endogenous *EHD2* expression, dCas9 Synergistic Activation Mediator Lentivector (pLenti-EF1a-dCas9-SAM, cat. # K015) and *EHD2* CRISPRa sgRNA Lentivector (pLenti-U6-sgRNA-PGK-Neo, cat. # K0663271) from Applied Biological Materials (Richmond, BC, Canada) were serially infected into MDA-MB-468 cell line. In all cases, clonal derivatives were obtained by limiting dilution and screened for complete KO using western blotting. Unless indicated, 3 or 4 clones (maintained separately) were pooled for experimental analyses.

## Cell lysates

Cells were lysed in RIPA (50 mM Tris pH 7.5, 150 mM NaCl, 1% Triton-X-100, 0.05% deoxycholate, 0.1% SDS, 1 mM phenylmethylsulfonyl fluoride (PMSF), 10 mM NaF, and 1 mM sodium orthovanadate) or Triton-X-100 (50 mM Tris pH 7.5, 150 mM NaCl, 0.5% Triton-X-100, 1 mM PMSF, 10 mM NaF, and 1 mM sodium orthovanadate) lysis buffer. Lysates were rocked at 4 °C for at least 1 hr, spun in a microfuge at 13,000 rpm for 20 min at 4 °C and supernatant protein concentration determined using the BCA (Thermo Fisher Scientific, Rockford, IL) or Bradford (Bio-Rad Laboratories, Hercules, CA) assay kits.

## Immunoprecipitation

One mg lysate protein aliquots were used for immunoprecipitation with optimized amounts of the indicated antibodies and 20 µL of Protein A Sepharose beads (GE Healthcare, Chalfont St. Giles, UK) followed by SDS-Polyacrylamide (Biorad) Gel Electrophoresis (PAGE), Polyvinylidene fluoride (PVDF) membrane (Bio-Rad Laboratories, Hercules, CA) transfer and Western Blotting, as described (*Luan et al., 2018*).

## Immunofluorescence microscopy

Immunofluorescence staining was performed as described (*Bailey et al., 2014*) with minor modifications. Cells cultured on glass coverslips were fixed with 4% PFA/PBS (10 min), blocked with 5% BSA/PBS (60 min), and incubated with primary antibodies in 5% BSA/PBS overnight at 4 °C. Coverslips were washed with PBS (3 x), incubated with the appropriate fluorochrome-conjugated secondary antibody for 45 min at room temperature (RT), washed and mounted using VECTASHEILD mounting

medium (cat. # H-1400, Vector Laboratories). For Structured Illumination Microscopy (SIM), images were acquired using a Zeiss ELYRA PS.1 microscope (Carl Zeiss). The Pearson coefficient of colocalization was determined using the FIJI package of ImageJ (NIH). For tissue staining, rehydrated tissue sections were boiled in antigen unmasking solution (H-3300, Vector Laboratories, Burlingame, CA) in a microwave for 20 min, slides were cooled, washed once in PBS, and blocked in heat-inactivated 10% FBS (SH30910.03, HyClone Laboratories, Logan, UT) for 1 h at RT. Primary antibodies diluted in blocking buffer were added overnight at 4 °C, slides were washed 3 x with PBS followed by incubation with Alexa Fluor 488 or 594-conjugated donkey anti-rabbit or anti-mouse secondary antibodies (Invitrogen, Carlsbad, CA) for 1 hr at RT in the dark. For negative controls, sections were incubated in the blocking buffer without the primary antibody. Nuclei were visualized with DAPI in antifade mounting medium (ProLong Gold Antifade mountant, Invitrogen, Carlsbad, CA). Fluorescence images were captured on a Zeiss LSM-710 confocal microscope.

## Proliferation assay

A Cell Titer-Glo assay was performed according to the manufacturer's specifications (Promega, Madison, WI).

## Extracellular matrix degradation assay

This assay was carried out using QCM Gelatin Invadopodia kit (Cat. # ECM670, EMD Millipore, Billerica, MA) according to the manufacturer's protocol. FITC-labeled gelatin was coated onto glass coverslips and crosslinked with 0.5% glutaraldehyde in PBS for 30 min. Coated coverslips were then washed 3 x each with PBS and 50 mM glycine in PBS. Cells were cultured for various time points to allow ECM degradation, seen as focal loss of fluorescent signal ('holes') in the labeled gelatin layer. The fluorescence intensity was further analyzed using the Image J software (NIH).

## Anchorage-independent growth assay

A total of 2500 cells were seeded in 0.35% soft agar on top of 0.6% soft agar layer in six-well plates. After 2 weeks, cells were stained with crystal violet and imaged under a phase contrast microscope. The number of colonies in the plate were enumerated using the Image J software (NIH). All experiments were done in triplicates and repeated three times.

## Tumor-sphere assay

Cells were re-suspended in DMEM/F12 media (# SH30023.01, GE Lifesciences) supplemented with 2 mM L-glutamine, 100 U/ml penicillin/streptomycin, 20 ng/ml EGF (sigma), 10 ng/ml FGF (R&D Systems) and 1 x B27 supplement (Gibco) and seeded at 100,000 cells/well in poly-HEMA coated six-well plates. After 1 week, tumor-spheres were imaged under a phase contrast microscope. Tumor-spheres greater than 40 μm in diameter were quantified as previously described (*Shaw et al., 2012*) using the Image J software (NIH, MD). All experiments were done in triplicates and repeated three times.

## Matrigel spheroid assay

Cells were re-suspended as a single cell suspension in media containing 50% Matrigel and seeded at 2000 cells/well on top of a base layer of Matrigel in an eight-well chamber slide. TNBC cell spheroids were allowed to grow for 7 days and then imaged and quantified. Quantification of invasive spheroids was performed by comparing the number of spheroids with invadopodia to the total number of spheroids. Over 100 total spheroids were counted per well and all experiments were done in triplicates and repeated three times.

## Trans-well migration and invasion assays

For migration assays, the cells were seeded in the top chambers of trans-wells (cat. # 353097, Corning) in serum-free medium. For invasion assay, the cells were seeded in Matrigel-precoated top chambers of trans-wells (cat. # 354480, Corning) in serum-free medium. Medium containing 10% FBS in the lower chamber served as a chemoattractant. After 18 hr, the cells on the upper side of the membrane were removed by scraping with cotton swabs, and cells on the lower side were fixed with methanol, stained with crystal violet, and counted. Experiments were run in triplicates and repeated three times.

## Orthotopic xenograft tumorigenesis

Three or five million cells in 0.1 ml 50% Matrigel (BD Biosciences) were implanted in the mammary fat pads of 4–6 weeks old non-pregnant female athymic nude mice (The Jackson Laboratory). Tumor growth was monitored using calipers weekly for up to 10 weeks. Tumor volume was calculated as length x width x depth/2 (*Luan et al., 2018*). Mice were euthanized when control tumors reached 2 cm$^3$ in volume or showed signs of ill health, as per institutional IACUC guidelines. At the end of the experiment the primary tumor, liver and lungs were resected, formalin-fixed and paraffin-embedded for further analyses.

## Analysis of tumor metastasis after tail vein injection of tumor cells

$10^6$ control or *EHD2*-KO MDA-MB-231 cells engineered with tdTomato/luciferase reporter were resuspended in 0.1 ml PBS and injected into the lateral tail-vein of 5-week-old non-pregnant female athymic nude mice. For bioluminescent imaging, mice received an intraperitoneal injection of 200 µl D-luciferin (15 mg/ml; cat. # L9504 from Millipore Sigma) 15 min before isoflurane anesthesia and were placed dorso-ventrally in the IVIS Imaging System (IVIS 2000). Images were acquired using the IVIS Spectrum CT and analyzed using Living Image 4.4 software (PerkinElmer). Mice were imaged weekly and followed for up to 40 days. At the end of the study, lungs were harvested from euthanized mice, fixed in paraformaldehyde, and embedded in paraffin for histopathological analysis.

## TIRF microscopy

Cells were seeded on 1.78 refractive index glass coverslips and transfected with pGFP-CAV1 (for CAV1 puncta) or STIM1-CFP (for STIM1 puncta). Cells were treated with or without thapsigargin (2.5 µM) before imaging. TIRF images were acquired using a TIRF video microscope (Nikon) equipped with CFI Apo TIRF 100A- NA 1.49 oil objective and an EMC CD camera (Photometrics HQ2). The surface CAV1 puncta were quantified using the ImageJ (NIH) software.

## Live-cell surface biotin labeling to assess the cell surface Orai1 levels

Cell monolayers were washed with ice-cold PBS, and incubated in the same buffer containing sulfo-NHS-LC-biotin (#A39257, Thermo Fisher) for 30 min at 4 °C. The cells were washed in PBS and their lysates in TX-100 lysis buffer subjected to anti-Orai1 immunoprecipitation followed by blotting with Streptavidin-Horseradish Peroxidase (HRP) Conjugate (cat. # SA10001) and enhanced chemiluminescence detection.

## Calcium flux assays

Cells were seeded in 35 mm glass-bottom dishes (cat. #FD35-100, WPI Inc) and loaded with Fluo4-AM in modified Tyrode's solution (2 mM calcium chloride, 1 mM magnesium chloride, 137 mM sodium chloride, 2.7 mM potassium chloride, 12 mM sodium bicarbonate, 0.2 mM sodium dihydrogen phosphate, 5.5 mM glucose, pH 7.4) for 1 hour. After washing with calcium-free Tyrode's solution, live cells were imaged under a confocal microscope (LSM710; Carl Zeiss), with fluorescence excitation at 488 nm and emission at 490–540 nm. To initiate the release of intracellular Ca$^{2+}$ stores, cells were stimulated with 2.5 µM thapsigargin in the absence of extracellular Ca$^{2+}$. Once the signals approached the baseline, calcium chloride was added to 2 mM final concentration to record the SOCE (*Lu et al., 2018*). Data are presented as fold change in fluorescence emission relative to baseline.

## Patient population and tissue microarrays

Tissue microarrays (TMAs) corresponding to a well-annotated 971 breast cancer patient cohort at the University of Nottingham Hospital Breast Unit were analyzed by IHC staining with a previously described anti-rabbit EHD2 antibody (*George et al., 2007*) that was further validated (*Figure 1A–B*). Of the whole series (840 cases), 759 were informative. Both cytoplasmic and nuclear EHD2 signals were recorded. For statistical analysis, the expression was dichotomized using cutoff points that were selected based on histogram distribution using the median and X-tile software as follows: a H-score of zero for nuclear EHD2 and H-score of 50 for cytoplasmic EHD2 expression. Statistical analysis was performed using the SPSS IBM 22 statistical software (SPSS Inc, Chicago, IL, USA). The relationship between nuclear and cytoplasmic EHD2 expression and different clinical-pathological variables was assessed using Chi square-test. Survival curves were obtained using Kaplan–Meier method for

outcome estimation and significance determined using the log-rank test. Two-tailed p-values less than 0.05 were considered significant. Multivariate analysis was performed using the Cox hazard analysis.

### Prognostic analysis and gene targeted correlation analysis of *EHD2*, *CAV1*, and *CAV2* mRNAs

The Kaplan–Meier plotter was used to evaluate the prognosis value of *EHD2*, *CAV1*, and *CAV2* mRNA expressions alone and in combination (*Györffy et al., 2010*). To analyze the survival probability alone and in combinations of *EHD2*, *CAV1*, and *CAV2* mRNA, the patient cohorts were split on the basis of trichotomization (T1 vs T3). The *EHD2* mRNA (probe set 221870_at), *CAV1* mRNA (probe set 212097_at), and *CAV2* mRNA (probe set 203323_at) were entered into the KM Plotter patient cohort basal-like (PAM50 subtype) patient cohort (n=953) and the relapse-free survival (RFS) was determined. The mean expression of *EHD2, CAV1* and *CAV2* were used to perform survival analysis of high *EHD2, CAV1* and *CAV2* vs. low *EHD2, CAV1* and *CAV2*. The hazard ratio (HR) with 95% confidence and log rank p-values were obtained from KM plotter. Gene correlation targeted analysis was performed to assess the correlation between *EHD2*, *CAV1* and CAV2 mRNA expression in basal-like (PAM50 subtype) breast cancer patients (n=783) and TNBC (IHC) cohort (n=293) using bc-GenExMiner v4.5 platforms. TCGA and SCAN-B RNAseq dataset were used. Correlation heatmap, correlation plots and Pearson's correlation coefficients computation were performed using bc-GenExMiner v4.5.

### Statistical analysis

Statistical analysis of in vitro data was performed by comparing groups using unpaired student's t test. In vivo tumorigenesis and metastasis data were analyzed using two-way ANOVA. A p value of<0.05 was considered significant.

### Human and animal subjects

Human tissues were collected and processed at the Nottingham University Hospital, United Kingdom. This study was approved by the Yorkshire & The Humber-Leeds East Research Ethics Committee (REC reference: 19/YH/0293) under the IRAS Project ID: 266925. Informed consent was obtained from all individuals prior to surgery for the use of their tissue materials in research. All samples were properly coded and anonymized in accordance with the approved protocols. All mouse xenograft and treatment studies were pre-approved by the UNMC Institutional Animal Care and Use Committee (IACUC) under the IACUC protocol number 19-115-10-FC and conducted strictly according to the pre-approved procedures, in compliance with Federal and State guidelines.

## Acknowledgements

We thank Drs. Anjana Rao for CFP-STIM1, Ari Helenius for Cav1-EGFP and Ian Frew for the pMule kit (used to assemble the tdTomato/luciferase) through Addgene. This research was funded by grants from DOD (W81XWH-17-1-0616 and W81XWH-20-1-0058 to HB and W81XWH-20-1-0546 to VB) and NIH (R21CA241055 and R03CA253193 to VB), and by Fred & Pamela Buffett Cancer Center (pilot grants to HB & VB) and the Raphael Bonita Memorial Fund. The UNMC core facilities are supported by the NCI Cancer Center Support Grant (P30CA036727) to Fred & Pamela Buffett Cancer Center and the Nebraska Research Initiative. TAB, AMB, IM and SC received University of Nebraska Medical Center Graduate Student Fellowships.

## Additional information

### Competing interests

Vimla Band, Hamid Band: received funding from Nimbus Therapeutics for an unrelated project. The other authors declare that no competing interests exist.

## Funding

| Funder | Grant reference number | Author |
|---|---|---|
| Department of Defence | W81XWH-17-1-0616 and W81XWH-20-1-0058 | Hamid Band |
| Department of Defence | W81XWH-20-1-0546 | Vimla Band |
| National Institutes of Health | R21CA241055 and R03CA253193 | Vimla Band |
| Fred & Pamela Buffett Cancer Center | Pilot grant | Vimla Band Hamid Band |
| University of Nebraska Medical Center | Graduate Student Fellowships | Timothy A Bielecki Aaqib M Bhat Sukanya Chakraborty Insha Mushtaq |
| Raphael Bonita Memorial Fund | | Hamid Band |

The funders had no role in study design, data collection and interpretation, or the decision to submit the work for publication.

## Author contributions

Haitao Luan, Conceptualization, Data curation, Formal analysis, Validation, Visualization, Methodology, Writing – original draft, Writing – review and editing; Timothy A Bielecki, Data curation, Formal analysis, Validation, Investigation, Visualization, Methodology, Writing – original draft, Writing – review and editing, Conceptualization; Bhopal C Mohapatra, Data curation, Formal analysis, Validation, Investigation, Visualization, Methodology, Writing – review and editing, Conceptualization; Namista Islam, Insha Mushtaq, Aaqib M Bhat, Sameer Mirza, Sukanya Chakraborty, Mohsin Raza, Michael S Toss, Wallace B Thoreson, Data curation, Formal analysis; Matthew D Storck, Resources, Project administration; Jane L Meza, Emad A Rakha, Formal analysis, Validation; Donald W Coulter, Formal analysis; Vimla Band, Conceptualization, Resources, Formal analysis, Supervision, Funding acquisition, Validation, Investigation, Methodology, Writing – review and editing; Hamid Band, Conceptualization, Resources, Data curation, Formal analysis, Supervision, Funding acquisition, Validation, Investigation, Visualization, Methodology, Writing – original draft, Writing – review and editing

## Author ORCIDs

Haitao Luan http://orcid.org/0000-0002-9831-167X
Wallace B Thoreson http://orcid.org/0000-0001-7104-042X
Vimla Band http://orcid.org/0000-0003-2014-7205
Hamid Band http://orcid.org/0000-0002-4996-9002

## Ethics

Human tissues were collected and processed at the Nottingham University Hospital, United Kingdom. This study was approved by the Yorkshire & The Humber-Leeds East Research Ethics Committee (REC reference: 19/YH/0293) under the IRAS Project ID: 266925. Informed consent was obtained from all individuals prior to surgery for the use of their tissue materials in research. All samples were properly coded and anonymized in accordance with the approved protocols.

All mouse xenograft and treatment studies were pre-approved by the UNMC Institutional Animal Care and Use Committee (IACUC) under the IACUC protocol number 19-115-10-FC and conducted strictly according to the pre-approved procedures, in compliance with Federal and State guidelines.

## Decision letter and Author response

Decision letter https://doi.org/10.7554/eLife.81288.sa1
Author response https://doi.org/10.7554/eLife.81288.sa2

# Additional files

## Supplementary files

• Supplementary file 1. Summary of EHD2 staining correlations in patients. (A) Validation summary

of EHD2 staining from all patients. (B) Associations between EHD2 nuclear and cytoplasmic expression and clinical, pathological, and biological characteristics in the complete patient series.

• MDAR checklist

## Data availability

All data generated or analysed during this study are included in the manuscript and supporting file.

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
