## [Editor Report]

This paper reports important findings regarding the role of EHD2 over-expression in a subset of poor-prognosis breast cancers. Through analysis of extensive patient data and in vitro and xenograft experiments with cell-line models, the paper provides solid evidence that EHD2, a component of caveolae, has strong pro-tumorigenic and pro-metastatic roles in these cancers by regulating calcium channels. The paper should be of interest to investigators studying metastasis and the role of caveolae in calcium signaling and homeostasis.

---

## [Decision Letter]

**Decision letter after peer review:**

Thank you for submitting your article "EHD2 overexpression promotes tumorigenesis and metastasis in triple-negative breast cancer by regulating store-operated calcium entry" for consideration by *eLife*. Your article has been reviewed by 2 peer reviewers, and the evaluation has been overseen by a Reviewing Editor and Jonathan Cooper as the Senior Editor. The reviewers have opted to remain anonymous. We apologize for the delay in the review process.

Essential revisions:

The reviewers agree that this is an interesting paper and most of the results justify the conclusions. The specific revisions are listed in the individual reviews below and many can be addressed by revising the text and tempering claims. The additional requested experiments do not appear to be major and would significantly improve the paper.

*Reviewer #2 (Recommendations for the authors):*

1. What is the function of EHD2 in nucleus? Does EHD2 shuffle between nucleus and cytoplasm? WB in Figure S1E on EHD2 between basal and luminal organoids can be strengthened.

2. Since EHD2 KO/KD inhibits tumor growth in vivo, the observation that EHD2 KO inhibits metastatic lesions could be attributed to tumor growth and may or may not be associated with metastasis.

3. Figure 5D. it is difficult to see if EHD2 and Cav1 are co-localized on plasma membrane. Current image looked like nuclear staining signals.

4. Figure 5: "These results support the conclusion that EHD2-dependent maintenance of cell surface caveolae is linked to its promotion of tumorigenic and pro-metastatic traits." This is under the assumption that CAV1's tumorigenic function is associated with cell membrane

5. Figure 6: If EHD2 specifically acts on SOCE, should EHD2 KO only inhibit the 2nd peak (after ca^2+^ addition) without changes on the 1st peak after Tg treatment? Current results reflects the possibility that EHD2 changes the structure of both plasma and ER membranes, resulting in decreased ca^2+^ trafficking. Where is EHD2 located?

6. Figure 7C. It is unclear which bands are Orai1.

7. Figure 8. In addition to inhibiting STIM1, SKF96365 also blocks TRPC channels, and is not a sole SOCE inhibitor.

---

## [Author Response]

Essential revisions:The reviewers agree that this is an interesting paper and most of the results justify the conclusions. The specific revisions are listed in the individual reviews below and many can be addressed by revising the text and tempering claims. The additional requested experiments do not appear to be major and would significantly improve the paper.Reviewer #2 (Recommendations for the authors):1. What is the function of EHD2 in nucleus? Does EHD2 shuffle between nucleus and cytoplasm? WB in Figure S1E on EHD2 between basal and luminal organoids can be strengthened.

The nuclear shuttling of EHD2 has been reported by others using cell line models (Pekar O, Benjamin S, Weidberg H, Smaldone S, Ramirez F, Horowitz M. EHD2 shuttles to the nucleus and represses transcription. Biochem J. 2012 Jun 15;444(3):383-94; Torrino S, Shen WW, Blouin CM, Mani SK, Viaris de Lesegno C, Bost P, Grassart A, Köster D, Valades-Cruz CA, Chambon V, Johannes L, Pierobon P, Soumelis V, Coirault C, Vassilopoulos S, Lamaze C. EHD2 is a mechanotransducer connecting caveolae dynamics with gene transcription. J Cell Biol. 2018 Dec 3;217(12):4092-4105), and these studies linked the nuclear translocation of EHD2 to repression or induction of certain genes (the latter in response to mechanical perturbations of cells), but the physiological or pathophysiological roles of nuclear EHD2 remain unknown. We have added appropriate statements discussing these findings in the Discussion section. Based on our findings, one could speculate the potential of the nuclear translocation as a mechanism to sequester EHD2 away from its membrane-proximal cell biological role in positively regulating the SOCE. We added this in the discussion but can remove it if it is deemed too speculative by the reviewers. To improve the blot presented as previous Figure S1E, we have conducted new experiments that more clearly show the higher expression of EHD2 in the basal compared to the luminal mammary epithelial cell-derived organoids (new Figure 1E).

2. Since EHD2 KO/KD inhibits tumor growth in vivo, the observation that EHD2 KO inhibits metastatic lesions could be attributed to tumor growth and may or may not be associated with metastasis.

We agree that detailed analyses of tumor cell extravasation and seeding will be needed to separate the role of reduced metastatic seeding versus growth in EHD2-dependent metastatic growth. We have added this caveat in the discussion.

3. Figure 5D. it is difficult to see if EHD2 and Cav1 are co-localized on plasma membrane. Current image looked like nuclear staining signals.

As indicated to a related comment of the Reviewer 1 for this experiment, we have re-done the EHD2-CAV1 colocalization analyses using Structured Illumination Microscopy (SIM) for higher resolution. As shown in new Figure 7D, the two proteins show strong co-localization at the plasma membrane and cytosol regions, and Pearson Correlation Coefficients show the co-localization to be significant in all three cell lines examined.

4. Figure 5: "These results support the conclusion that EHD2-dependent maintenance of cell surface caveolae is linked to its promotion of tumorigenic and pro-metastatic traits." This is under the assumption that CAV1's tumorigenic function is associated with cell membrane

We agree that our statement did not consider the potential role of CAV1 and caveolae at subcellular locations other than the plasma membrane. We have modified our conclusion with this caveat that future studies will need to address.

5. Figure 6: If EHD2 specifically acts on SOCE, should EHD2 KO only inhibit the 2nd peak (after ca^2+^ addition) without changes on the 1st peak after Tg treatment? Current results reflects the possibility that EHD2 changes the structure of both plasma and ER membranes, resulting in decreased ca^2+^ trafficking. Where is EHD2 located?

We appreciate the reviewer’s insight and have discussed this in more detail. In essence, the SOCE is known to be a critical mechanism for refilling of intracellular stores, especially the Endoplasmic Reticulum (ER) (Chung WY, Jha A, Ahuja M, Muallem S. Ca(2+) influx at the ER/PM junctions. Cell Calcium 2017;63:29-32). Thus, a defect in SOCE is expected to also reduce the amount of ca^2+^ in intracellular stores and hence a smaller first peak upon SERCA2 inhibition. This can be also seen when cells are treated with SOCE inhibitors. Consistent with this interpretation, use of membrane targeted reporter shows that in fact the SOCE is inhibited in EHD2-KO cells (new Figure 8F). However, we agree with the Reviewer’s caution that EHD2 may also act at the level of intracellular stores since localization analyses show its presence in intracellular structures besides the plasma membrane. We have added this cautionary note to the Discussion.

6. Figure 7C. It is unclear which bands are Orai1.

We have indicated the Orai1 bands in the figure. Depending on the level of exposure (and how much is present in the lysates or immunoprecipitates), Orai1 is seen as a smear or a series of more discrete bands with some smear (likely reflecting its multi-pass transmembrane nature and potential modifications) (See also new Figure 9—figure supplement 1).

7. Figure 8. In addition to inhibiting STIM1, SKF96365 also blocks TRPC channels, and is not a sole SOCE inhibitor.

We used the SKF96365 compound because of its extensive prior use as an SOCE inhibitor but we agree with the reviewer and have noted this caveat in our discussion with references to its broader specificity.